# Quantitative structured illumination microscopy via a physical model-based background filtering algorithm reveals actin dynamics

Yanquan Mo [1], Kunhao Wang[2], Liuju Li[1], Shijia Xing[1], Shouhua Ye[3], Jiayuan Wen[3], Xinxin Duan[4], Ziying Luo[3], Wen Gou[5], Tongsheng Chen [2], Yu-Hui Zhang [4], Changliang Guo [1], Junchao Fan [5] ✉ & Liangyi Chen [1,6,7,8] ✉

Despite the prevalence of superresolution (SR) microscopy, quantitative live-cell SR imaging that maintains the completeness of delicate structures and the linearity of fluorescence signals remains an uncharted territory. Structured illumination microscopy (SIM) is the ideal tool for live-cell SR imaging. However, it suffers from an out-of-focus background that leads to reconstruction artifacts. Previous post hoc background suppression methods are prone to human bias, fail at densely labeled structures, and are nonlinear. Here, we propose a physical model-based Background Filtering method for living cell SR imaging combined with the 2D-SIM reconstruction procedure (BF-SIM). BF-SIM helps preserve intricate and weak structures down to sub-70 nm resolution while maintaining signal linearity, which allows for the discovery of dynamic actin structures that, to the best of our knowledge, have not been previously monitored.

The emergence of superresolution (SR) microscopy rapidly transforms different life science disciplines, such as cell biology. However, existing SR methods are used mainly for structural imaging, while their applications in quantitative live-cell SR studies remain limited. SR methods based on single molecules are either slow (based on individual localizations)[1,2] or nonlinear (based on ensemble fluorescence fluctuations)[3,4]. Stimulated emission depletion microscopy, on the other hand, requires extensive illumination power to obtain images with good signal-to-noise ratios (SNRs), which often leads to continuous photo-bleaching and incomplete structures achievable[5,6].

Due to its superior photon efficiency, SIM has been the method of choice for live-cell SR imaging[7]. Many variants have been proposed to reduce phototoxicity further and improve image contrast and resolution, including TIRF-SIM[8], GI-SIM[9], Hessian-SIM[10], and Sparse-SIM[11]. Despite these achievements, as a computational method, SIM is prone to reconstruction artifacts[12,13]. For raw images with low SNRs, people can evaluate and control the noise during reconstruction using noise-dependent reconstruction parameters[14]. We use prior knowledge of the spatiotemporal continuity of genuine signals to develop Hessian-SIM, which suppresses artifacts due to amplified random noise upon deconvolution[10]. However, as a widefield method, SIM also suffers

[1]State Key Laboratory of Membrane Biology, Center for Life Sciences, College of Future Technology, Peking University, Beijing 100871, China. [2]Key Laboratory of Laser Life Science, Ministry of Education, College of Biophotonics, South China Normal University, Guangzhou 510631, China. [3]Guangzhou Computational Super-resolution Biotech Co., Ltd, Guangzhou 510535, China. [4]Britton Chance Center and MOE Key Laboratory for Biomedical Photonics, Wuhan National Laboratory for Optoelectronics, Huazhong University of Science and Technology, Wuhan 430074, China. [5]Chongqing Key Laboratory of Image Cognition, College of Computer Science and Technology, Chongqing University of Posts and Telecommunications, Chongqing 400065, China. [6]PKU-IDG/McGovern Institute for Brain Research, Beijing 100871, China. [7]Beijing Academy of Artificial Intelligence, Beijing 100871, China. [8]National Biomedical Imaging Center, Beijing 100871, China. ✉e-mail: fanjc@cqupt.edu.cn; lychen@pku.edu.cn

from an out-of-focus background that reduces image contrast and generates hammer-stroke and honeycomb artifacts[12]. The spatio-temporal continuity constraint Hessian algorithm cannot resolve these fixed patterns of artifacts[10]. Alternatively, people have proposed parameter-selective frequency spectrum attenuation methods to suppress these artifacts, such as Notch filtering[15], HiFi-SIM[16], and JSFR-SIM[17]. However, these post hoc methods require parameter tuning prone to human bias and fail at densely labeled fluorescent structures. Most importantly, because these methods do not handle the physical existence of out-of-focus fluorescence per se, they may nonlinearly change the amplitudes of reconstructed fluorescence signals. Therefore, quantitative SR imaging and analysis of live-cell dynamics remain a challenge to be addressed.

With over 100 binding partners, actin and actin-related structures profoundly regulate all aspects of cellular functions, such as cell shape, mechanics, division, and migration[18]. In addition to well-known architectures like lamellipodium, filopodium, stress fibers, and cortex, many transient actin structures have been reported recently, such as actin waves[19], moving patches[20], arcs[21], flashes[22], and tentpoles[23]. Therefore, live cells use the non-equilibrium state of actin to perform its localized and dynamic function. With meshed actin pores with diameters down to 50 nm[24,25], no current SR microscopy can visualize details of these intricate structural dynamics in live cells[24,25]. Therefore, researchers rely on image cross-correlations and optical flow visualization for their quantitative analysis[26–29]. However, from these analyses, the accuracy of various biophysical models proposed to recapitulate the cellular process in vivo remains unclear[18,30,31].

Here we propose a physical model-based Background Filtering (BF) method, which removes the background with minimal impacting the information. Combined with different SIM reconstruction algorithms, it improves the fidelity and integrity of intricate structures resolved. With a sub-70 nm resolution conferred by the BF-Sparse-SIM, we show accurate tracking of actin waves of the lamellipodium in macrophages. In addition, we characterize three additional types of localized cortical actin dynamics, which may enable more insights and better modeling of actin's dynamic assembly and disassembly.

## Results

By analyzing the honeycomb artifacts, we found that artifacts manifested in SIM when the reconstructed SR spectrum showed apparent high-frequency patches, which resulted from the incomplete separation and shift of the frequency spectrum of background fluorescence[12,16]. As the SIM background originated from the defocused emitters, we built a realistic physical model approximating the out-of-focus fluorescence by integrating emission over the corresponding depths of the three-dimensional point spread function (3D PSF) (Fig. 1a). Therefore, we rewrote the imaging model of the SIM as:

$$d(\mathbf{r}) = [g(\mathbf{r}) \cdot I(\mathbf{r})] \otimes h(\mathbf{r},z_{in}) + [g(\mathbf{r}) \cdot I(\mathbf{r})] \otimes h(\mathbf{r},z_{out}). \quad (1)$$

where $d(\mathbf{r})$ is the image detected by the camera, $I(\mathbf{r})$ is the sinusoidal illumination pattern that is the same at different depths of 2D images, $g(\mathbf{r})$ is the sample distribution that we assume to remain stable during raw image acquisition, and $h(\mathbf{r},z_{in})$ and $h(\mathbf{r},z_{out})$ are the PSFs within the range in-focus $z_{in}$ and with the range out-of-focus $z_{out}$, respectively. $\mathbf{r}$ is the 2D space position vector and $\otimes$ is the convolution operator.

Hence, the in-focal signal distribution is equal to the original raw image minus the defocused background, and according to Eq. (1):

$$g(\mathbf{r}) \cdot I(\mathbf{r}) = \mathrm{iFt}\left\{ \frac{\mathrm{Ft}\{d(\mathbf{r})\}}{\mathrm{Ft}\{h(\mathbf{r},z_{in})\} + \mathrm{Ft}\{h(\mathbf{r},z_{out})\}} \right\} = \mathrm{iFt}\left\{ \frac{D(\mathbf{k})}{H_{in}(\mathbf{k}) + H_{out}(\mathbf{k})} \right\}, \quad (2)$$

$$d_{in}(\mathbf{r}) = d(\mathbf{r}) - [g(\mathbf{r}) \cdot I(\mathbf{r})] \otimes h(\mathbf{r},z_{out}) = d(\mathbf{r}) - \mathrm{iFt}\left\{ D(\mathbf{k}) \cdot \frac{H_{out}(\mathbf{k})}{H_{in}(\mathbf{k}) + H_{out}(\mathbf{k})} \right\}. \quad (3)$$

Therefore, we used the $d_{in}(\mathbf{r})$ that removes out-of-focus fluorescence for subsequent SR reconstruction. Instead of experimentally measuring the corresponding $H_{out}(\mathbf{k})$ and $H_{in}(\mathbf{k})$, we used the simulated 3D PSF built from the PSF Generator plugin in ImageJ[32] according to the wavelength and numerical aperture. We considered signals within the axial resolution range as in-focus and those outside the resolution range as out-of-focus background[33]. For a high numerical aperture system[10], we regarded the PSF within the range of $-0.4$–$0.4\,\mu m$ as $h(\mathbf{r},z_{in})$ and that in the range of $-4$–$-0.4\,\mu m$ and $0.4$–$4\,\mu m$ as $h(\mathbf{r},z_{out})$. Thus, we named the pipeline Background-Filtered SIM reconstruction (BF-SIM), while the detailed calculation process was given in Supplementary Note 1 and Supplementary Figs. 1–3.

### BF-SIM offers the optimal tradeoff between background suppression and weak signal retention

First, we tested the performance of BF-SIM on fluorescently labeled actin, which was distributed all over the cell and produced uneven and excessive background. After frequency unmixing and shifting, six prominent spots in the spectrum emerged, which caused artifacts after the conventional Wiener restoration (Fig. 1b,c). We used standard methods, such as the rolling ball and wavelets, to remove the background from the raw images before the subsequent frequency shifting and Wiener reconstruction. While they generated final SR images of improved contrast, weak signals were often removed (white arrows, Supplementary Fig. 4). While the HiFi-SIM algorithm attempted to resolve this problem by designing OTF attenuation (Gaussian filtering) at the corresponding positions of high- and low-frequency components[16], selecting the correct attStrength (the Gaussian filter parameter) is challenging. At an attStrength too small (0.2), the HiFi-SIM failed to suppress the background; thus, some faint stripe artifacts persisted in the final result (red arrows, Supplementary Fig. 4); at an attStrength too large (0.99), the background suppression was overdone and again removed weak signals (Supplementary Fig. 4). After trial-and-error for several rounds, we found the appropriate attStrength (0.92) that suppressed the background without removing many signals from the SR reconstruction. In addition, we have also compared BF-SIM side-by-side with the notch filtering method (NF-SIM)[34], which lacks the second deconvolution step to compensate for removing weak signals compared to the HiFi-SIM (Supplementary Fig. 5, 6). We found that NF-SIM reconstruction was sensitive to parameter selection, as a fluctuation of attStrength of 0.01 led to significantly different results (Supplementary Fig. 6). Besides, an attStrength of 0.999 was sufficient for NF-SIM to suppress the background at the price of removing weak actin filaments (Supplementary Fig. 6). However, such a parameter was not ineffective in eliminating residual artifacts in the NF-SIM reconstruction of the fluorescent beads (Supplementary Fig. 5). Therefore, BF-SIM excels in suppressing background and retaining weak signals and is superior in algorithm robustness and accessibility.

Overall, our BF-SIM removed out-of-focus fluorescence more efficiently, as seen in the much-reduced spots in the spectrum after frequency shifting (Fig. 1b,c, Movie 1). Consequently, the average contrast of actin filaments under BF-SIM was more than 200% under Wiener-SIM and 130% under HiFi-SIM (Fig. 1e). The effect was even more dramatic in filaments with low SNR, as a weak branch was not detected under either Wiener-SIM or HiFi-SIM but became apparent with BF-SIM (Fig. 1d). Time-lapse imaging confirmed the actual existence of the branch. In contrast to the marginal detection of the signal at times $t_1$ and $t_4$ by HiFi-SIM (red arrows, Supplementary Fig. 7), BF-SIM robustly resolved the filament at all time points. The

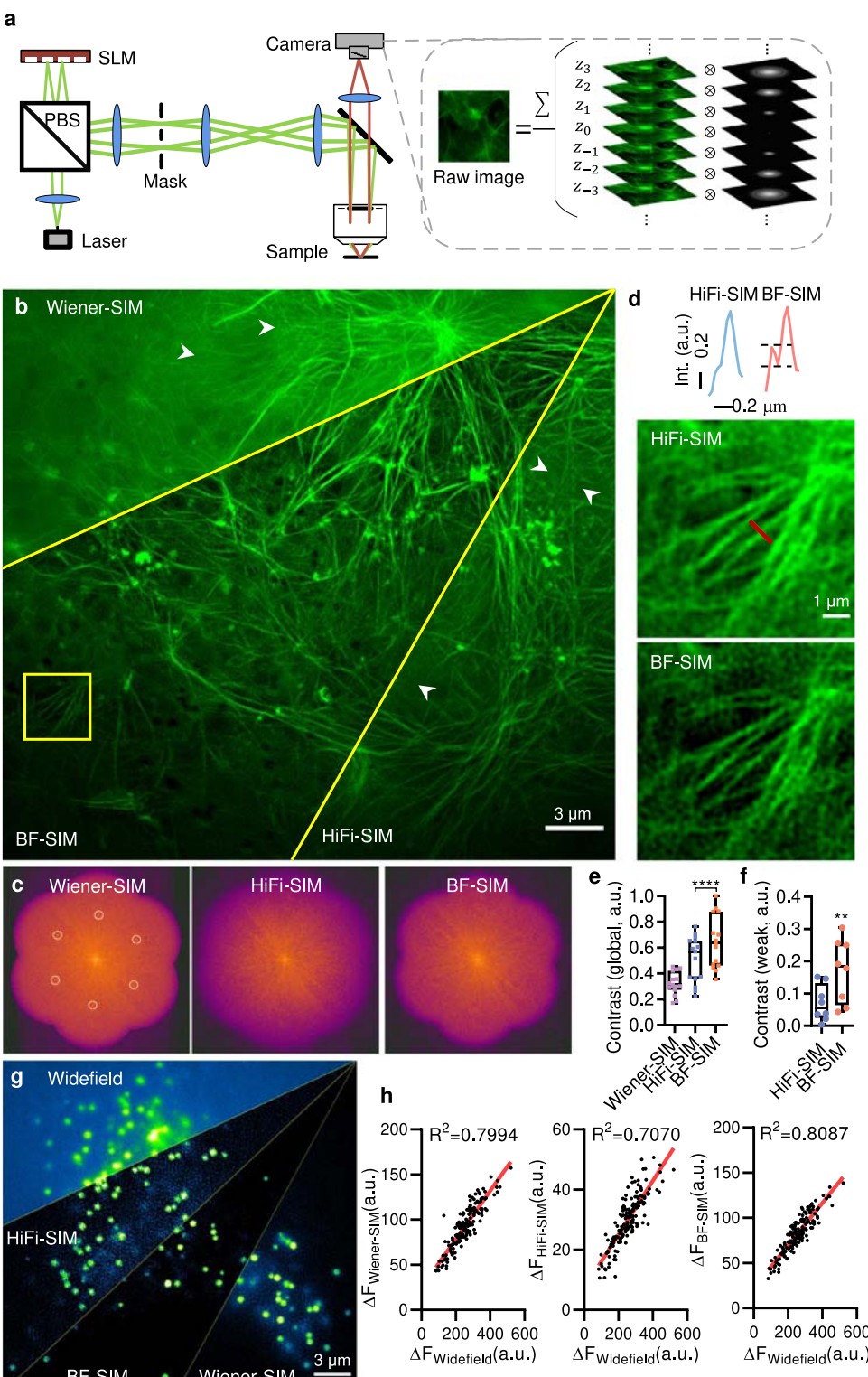

average contrast of weakly labeled filaments under BF-SIM was twofold of that under HiFi-SIM (Fig. 1f). By comparing the fluorescence intensities of lipid droplets measured under widefield, Wiener-SIM, HiFi-SIM, and BF-SIM, we quantitatively evaluated the signal linearities after SR reconstruction. Like Wiener-SIM, BF-SIM achieved a higher linear correlation with widefield measurements than HiFi-SIM (Fig. 1g,h).

Realistic background filtering also helped improve SR reconstruction of multilayer two-beam interference SIM data. In reconstructing layer-by-layer 3D tubulin filaments in fixed cells, BF-SIM

demonstrated better optical section capability than HiFi-SIM (Supplementary Fig. 8, Movie 2). These data established the superiority of our method, which offers an optimal tradeoff between background suppression, artifact removal, and weak signal retention without parameter adjustment in live-cell SR volumetric imaging.

## BF enhances the fidelity and integrity of intricate structures reconstructed by other SIM algorithms

Many open-source software programs based on conventional Wiener reconstruction exist, including SIMtoolbox[35], openSIM[36], and fairSIM[34].

**Fig. 1 | The principle of the BF method and its superior performance compared with other background suppression algorithms. a** Schematic illustration of the 2D-SIM setup and the principle of background generation of the captured image. On the left is a diagram of the light path. Blue oval: Lens; PBS: polarization beam splitter; SLM: spatial light modulator. The dashed box on the right is the schematic diagram of the source of the raw image collected by the camera, which includes the signals by the in-focal PSF convoluted image and the background by the out-of-focal PSF convoluted image. **b** Representative example of a COS-7-cell labeled with LifeAct-EGFP and reconstructed by traditional Wiener-SIM, HiFi-SIM, and BF-SIM. **c** Corresponding reconstructed spectrum. Compared with Wiener-SIM, HiFi-SIM, and BF-SIM can suppress high-frequency patchy features in the spatial frequency spectrum. **d** Enlargement of the ROI in the yellow box in (**b**), the SR image of HiFi-SIM and BF-SIM, and the normalized intensity profiles along the red line in HiFi-SIM and BF-SIM. As shown by the dotted line, the weak signal of BF-SIM is not affected. **e** Normalized contrast ratio of the global image of three reconstruction algorithms ($n = 5$ biologically independent cells $\times$ 3 frames). **f**, Normalized contrast ratio of the weak local signal. BF-SIM is superior to HiFi-SIM in both global contrast and weak signal contrast ($n = 8$ filaments from Fig. 1b), $p = 0.0027$ (BF-SIM vs HiFi-SIM). **g** Widefield, Wiener-SIM, HiFi-SIM, and BF-SIM SR images of the lipid droplets (LipidSpot 488) of living COS-7 cells. **h** Correlations between the peak fluorescence intensities of lipid droplets under Wiener-SIM (left), HiFi-SIM (middle), and BF-SIM (right) and those under a widefield microscope ($n = 165$ lipid droplets from 4 cells). The fluorescence intensities obtained with BF-SIM demonstrate a higher correlation than those obtained with HiFi-SIM. We used the two-tailed paired Student's t-test for data in (**e,f**). **$p < 0.01$, ***$p < 0.001$.

Using fairSIM as an example, we tested whether the BF pipeline could help them. From living mitochondria data obtained under 2D-SIM, fairSIM produced severe honeycomb and hammer-stroke artifacts, and some cristae structures diminished in the background (Supplementary Fig. 9). Adding the BF preprocessing step improved the subsequent reconstruction quality, resulting in a reduced background, an ~30% increase in contrast, and a better appreciation of mitochondrial cristae structures (Supplementary Fig. 9, Movie 3).

To reduce the photon budgets required for SIM and artifacts, researchers have developed end-to-end deep-learning networks, such as GAN[37], DL-SIM[38], and DFCAN[39], to perform SIM reconstructions. However, these state-of-the-art deep-learning networks still suffer from densely labeled samples with a strong background, as shown in the actin reconstructed by DFCAN (Fig. 2a–d). If the raw images were preprocessed with the BF procedure followed by DFCAN, actin filaments showed an average contrast increase of ~60%, while weak branches were also better resolved (Fig. 2a–d).

Then, we tested the BF procedure on our recently developed sparse deconvolution algorithm that enables computational SR. Based on the spatiotemporal continuity and relative sparsity of fluorescence signals, the algorithm extends resolution by performing constrained iterative optimizations followed by iterative RL deconvolution[11]. However, we noticed that increases in resolution sometimes come at the price of possible enhanced artifacts and weak signal removal[11]. For example, fix-patterned artifacts in SIM images also satisfy the continuity and sparsity a prior; thus, they are reinforced by sparse deconvolution. Indeed, weak periodic honeycomb artifacts of mitochondria and endoplasmic reticulum images became prominent after sparse deconvolution. However, these artifacts were removed if the SIM images were pretreated with the BF method before deconvolution (BF-Sparse-SIM, Supplementary Fig. 10, Movies 4, 5). Despite increased resolution from $124.0 \pm 0.8$ nm to sub-70 nm (indicated by FWHMs of filaments $61.5 \pm 0.5$ nm in Fig. 2g, minimum FRC values $66.8 \pm 0.3$ nm and $67.0 \pm 0.4$ nm by PANELJ[40] and NanoJ-SQUIRREL[41], respectively; mean FRC $78.9 \pm 1.7$ nm by NanoJ-SQUIRREL, Supplementary Fig. 11d,e), some weak fluorescent filaments in dense actin meshes were invisible after sparse deconvolution using the original wavelets background suppression method[11], while strong and long filaments became discontinuous and intermittent (Fig. 2e,f, Movie 6). The background removal by the BF pretreatment suppressed these side effects again, highlighted by the 2-3-fold fewer variations in fluorescence intensities along bright and weak filaments while retaining the resolution (Fig. 2g,h, Supplementary Fig. 11d). Subsequently, we observed more actin filaments after BF pretreatment (~147% increase in density), and the average length of actin filaments was nearly threefold longer (Fig. 2e–i, Supplementary Fig. 11).

### Accurate analysis of actin waves and revelation of actin dynamics by BF-Sparse-SIM

With pore diameters of 50–200 nm, concentrated actin forms a gel-like, dynamic network under the lamellipodium and the cell cortex[24,25], which are challenging tasks for previous live-cell SR methods. With sub-70 nm resolution and the ability to resolve weakly-labeled actin filaments in live cells, we used BF-Sparse-SIM to examine the dynamics of traveling actin waves in macrophages. In a living RAW264.7 cell expressing LifeAct-EGFP, we observed rapid actin retracting from the cell periphery to the cell center, resembling traveling waves (Fig. 3a). Using the published software called spatiotemporal image correlation spectroscopy (STICS)[26,27], we analyzed the speed and direction of actin flow (Fig. 3b). Compared with results from images obtained by other algorithms, BF-Sparse-SIM showed more consistent flow directions between different frames, which agreed with the visually observed process (Fig. 3c). In addition, the mean velocity calculated from BF-Sparse-SIM reconstruction was the highest among all reconstructions (Fig. 3d), confirming the reduced mean speeds due to spatial downsampling. Therefore, enhancing resolution and removing confounding backgrounds improve the accuracy of analytic results.

In addition, we identified three types of localized actin dynamics in the cell cortex. Individual small actin punctum emerged at the network node before disassembling immediately after its fluorescence peaked (Fig. 4a,b,h); thus, we termed it an "actin blip". On the other hand, one actin punctum was followed by the appearance of more actin puncta in the proximity, representing collaborative actin assembly before slowly spreading over the cellular footprint laterally (Fig. 4c,d,h). Since it demonstrated a much increased fluorescence-labeled field of view and slightly reduced mean peak fluorescence intensity than the actin blip, we named the process "actin cloud". Finally, we occasionally observed the sequential emergence of fluorescence puncta forming ring-like structures with a mean diameter of $0.93 \pm 0.07$ μm (9 events from 3 cells), followed by the spiral brightening within the circle, termed "actin vortex" (Fig. 4e,f,h, Supplementary Fig. 12, Movie 7). Intriguingly, half of the actin vortex events were accompanied by another rapid increase in fluorescence intensity before laterally diffusing away (Fig. 4g). This resulted in a much higher mean peak fluorescence intensity of the actin vortex than the other two types (Fig. 4h). We also calculated kinetics of these transient actin dynamics under the conventional Wiener-SIM and BF-SIM, which demonstrated similar trends to those obtained by the BF-Sparse-SIM (Supplementary Fig. 13 and Supplementary Table 1). In addition, we have imaged with spinning-disc structured illumination microscopy (SD-SIM)[42] and stimulated emission depletion (STED)[6] microscopy, which validated the existence of actin blips, clouds, and vortexes in live RAW264.7 cells (Supplementary Figs. 14, 15 and Supplementary Table 2). Therefore, the revelation of these transient and localized actin dynamics in macrophages reinforces the importance of actin in regulating various aspects of cellular function and the necessity of quantitative SR methods to study them in live cells.

## Discussion

Here we established the superiority of the model-based BF as a preprocessing step before subsequent SIM reconstructions for live cell SR imaging. Although from the perspective of suppressing low-frequency background components, the effect of BF is equivalent to attenuating the spectrum in Wiener-SIM. However, unlike the methods of directly

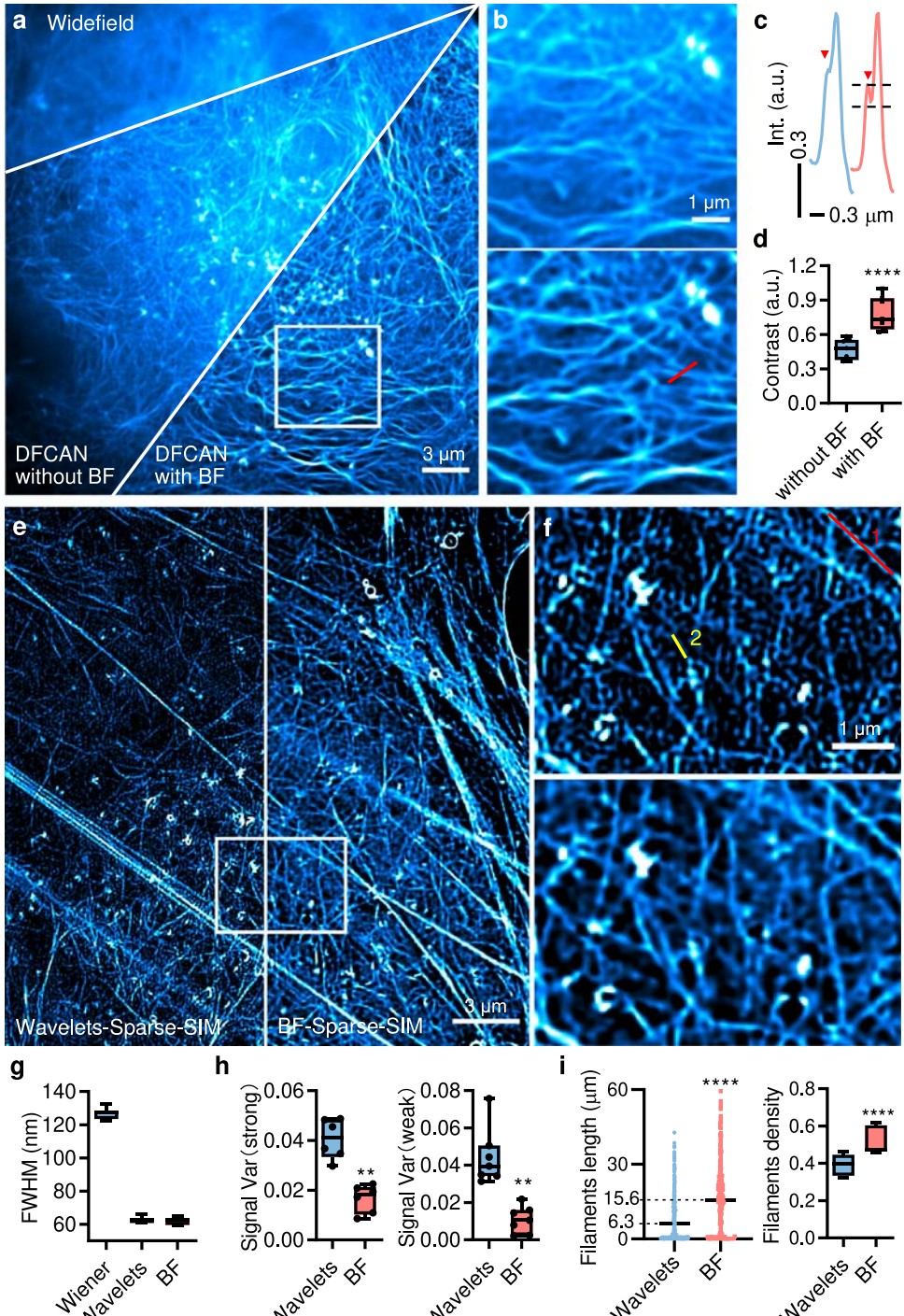

**Fig. 2 | Adding the BF step to other SIM reconstruction algorithms improves image contrast, while BF-Sparse-SIM achieves high resolution and retains the complete intricate, dense structures. a** The widefield image of LifeAct-EGFP in a living COS-7 cell and corresponding SR images were reconstructed by DFCAN without or with BF preprocessing. **b** ROI enlargement in the white box in (**a**). **c** Normalized intensity profiles along the red line in (**b**). As the red arrows show, the BF improves the visibility of weak signals. **d** Normalized contrast ratio of SR images without and with BF, and the latter enhances the contrast of SR images predicted by DFCAN (*n* = 7 biologically independent cells). **e** The actin filaments (LifeAct-EGFP) of a living U2OS cell under Sparse-SIM with the original wavelets background suppression or BF preprocessing. **f** ROI enlargement in the white box in (**e**). **g** FHWMs of filaments under the Wiener-SIM, Sparse SIM with either wavelets, or BF

background subtraction pretreatments (42 filaments from 3 cells). **h** Average signal variances in fluorescence intensities along actin filaments were strongly (*n* = 6 filaments from Fig. 2e, *p* = 0.0043) or weakly (*n* = 7 filaments from Fig. 2e, *p* = 0.0011) labeled. The Red 1 is strong, while the Yellow 2 is faint in (**f**). **i** Length of actin filaments after segmentation and skeletonization (*n* = 3 biologically independent cells) and density of filaments after segmentation (*n* = 3 biologically independent cells × 5 frames). The average length of actin filaments with the wavelets and the BF step was 6.3 ± 0.23 μm and 15.6 ± 0.73 μm, respectively. Adding BF preprocessing to Sparse-SIM enhanced signal continuity, avoided losing weak signals, and improved signal fidelity. We used the two-tailed paired Student's *t*-test for data in (**d,h**) and the two-tailed unpaired Student's *t*-test for data in (**i**). **\*\***p<0.01, **\*\*\*\***p<0.0001.

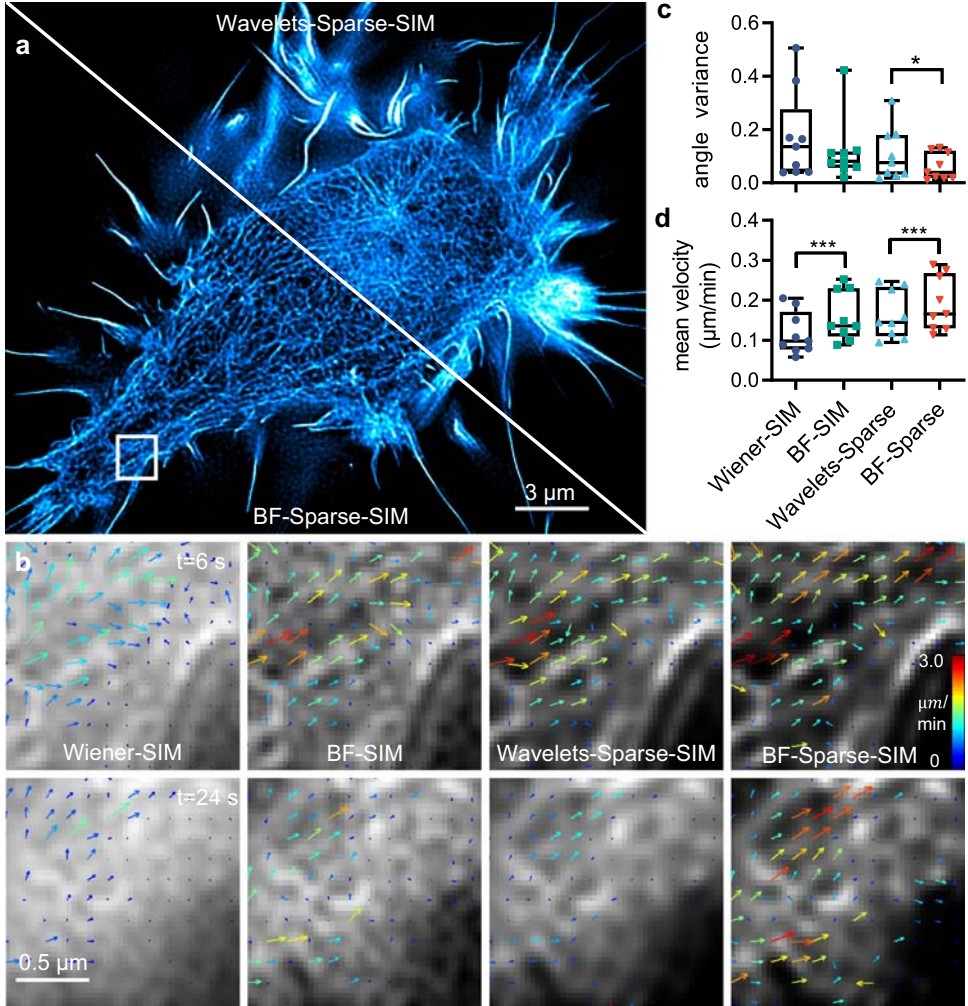

**Fig. 3 | More accurate and quantitative analysis of actin waves in macrophages observed under the BF-Sparse-SIM. a** Actin filaments within a representative living RAW264.7 cell (labeled as lifeAct-EGFP) under the Sparse-SIM with the original wavelets background suppression or BF preprocessing. **b** The velocity distribution of the ROI region in (**a**) when $t = 6$ s and $t = 24$ s. **c** The variance of actin's main flow direction (the average velocity angle with an amplitude more significant than 60 deciles) between different frames ($n = 9$ ROIs from 3 biologically independent cells), $p = 0.0321$ (Wavelets-Sparse-SIM vs BF-Sparse-SIM). **d** The average actin wave speeds under the Wiener-SIM, BF-SIM, and Sparse-SIM with either wavelets, or BF background subtraction pretreatments ($n = 9$ ROIs from three biologically independent cells), $p = 0.0001$ (Wiener-SIM vs BF -SIM) and $p = 0.0003$ (Wavelets-Sparse-SIM vs BF-Sparse-SIM). We used the two-tailed paired Student's $t$-test for data in (**c,d**). *$p<0.05$, ***$p<0.001$.

parameter-selective attenuating particular regions of the spectrum, our BF method is inspired by the physical model of the imaging process which is free of parameter tuning for different samples. Essentially, the process of selecting parameters for different organelles using the traditional artificial attenuation methods resembles probabilistic modeling where a theoretical distribution is fitted to available data. However, Kiureghian et al. have pointed out that conventional goodness-of-fit tests do not ensure accurate fitting in the tail region, where rare events are likely to occur, when probabilistic models are used[43].Consequently, it is challenging to estimate the magnitude of errors that arise from the selection of a distribution model, which sharply differs from the physical model. Therefore, this highlights a fundamental difference that distinguishes our BF method from traditional spectrum attenuation methods and future learning-based background-removing methods. While all these methods can suppress fixed pattern artifacts of SIM images due to high background fluorescence, the BF method better maintains the linearity among weak and strong signals after reconstruction as it models realistic emitters. Although all the background-suppressed methods may remove some signals from the unbound cytoplasmic protein pool, we

showed that $Ca^{2+}$ transients under the BF-SIM are highly correlated to those under the widefield microscopy, achieved a correlation level similar to the Wiener-SIM (Supplementary Fig. 16). This data agrees with the superiority of the BF-SIM in maintaining signal linearity over other background suppressing methods, as demonstrated by the comparison of lipid droplets of different fluorescent intensities (Figs. 1g and 1h), and good Resolution scale error[41] values by the BF-SIM reconstructions for actin filaments and fluorescent beads (Supplementary Fig. 5, 6). Applying the BF method to plane-by-plane images obtained by the two-beam interference SIM generated good volumetric SR reconstructions. Because two-beam interference SIM takes 6 frames less than three-beam interference SIM per plane, it will lead to less photobleaching and phototoxicity. Therefore, two-beam BF-Sparse-SIM may offer an optimal compromise among photon dosage, background fluorescence intensity, spatial resolution, and the completeness of the structure to be imaged.

Combined with the sparse deconvolution, BF-Sparse-SIM achieves sub-70 nm resolution without compromising image contrasts and weak structures. These enable the dynamics of actin waves, one of the more dense subcellular structures to be visualized and quantified in

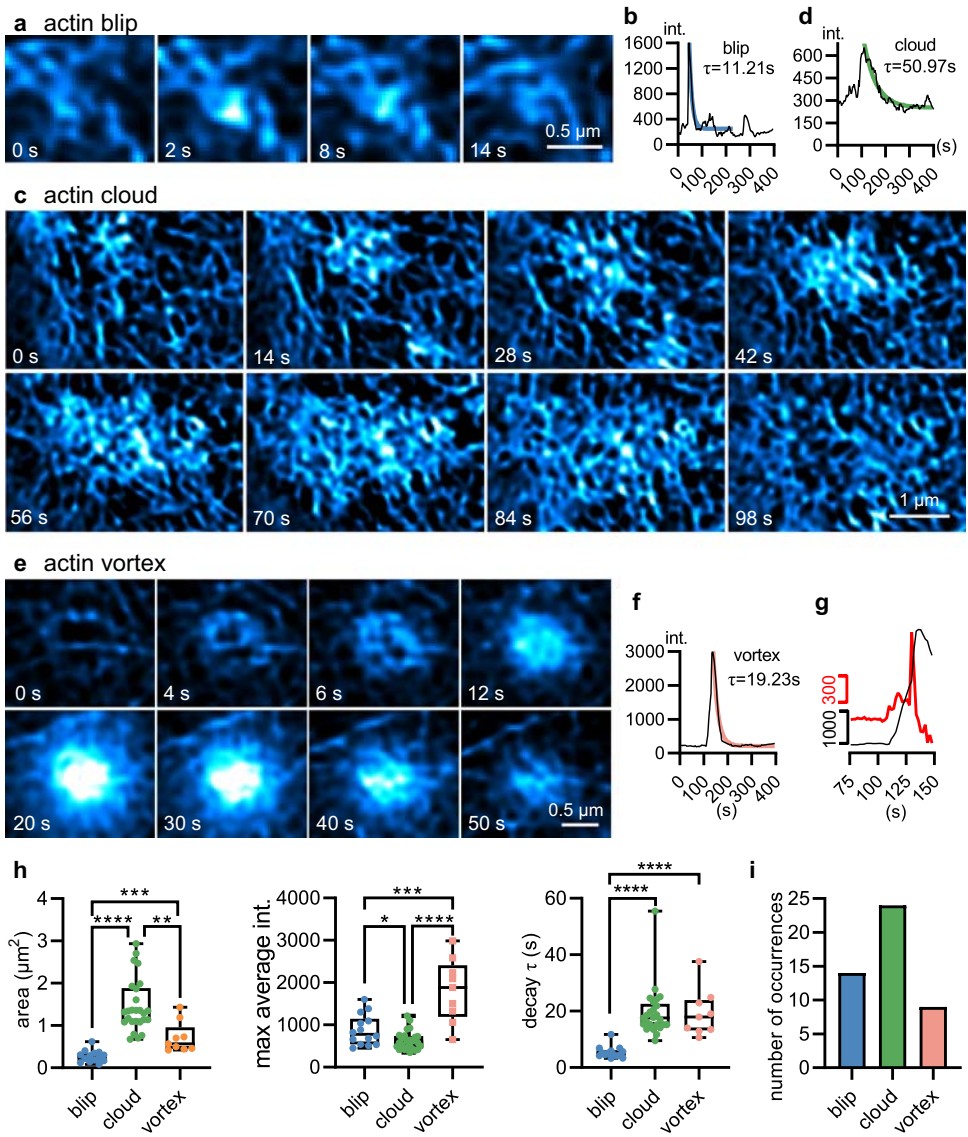

**Fig. 4 | Three types of localized transient actin dynamics.** A representative example of an actin blip event that was small in size (**a**). It demonstrated a rapid increase and decay in fluorescence intensity (**b**). A representative example of an actin cloud event that was large (**c**). It demonstrated a slow increase, an intermediate plateau, and the final slow decay in fluorescence intensity (**d**). A representative example of an actin vortex event that was medium in size, and increased fluorescence intensities followed the spiral and outward-to-inward pattern (**e**). It demonstrated an immediate increase and decay in fluorescence intensity (**f**). In this event, we observed two apparent speeds of intensity increases from the original intensity trace (gray) and the differentiated intensity trace (red) (**g**). **h** The sizes (left, blip $0.270 \pm 0.037$ μm$^2$,

cloud $1.48 \pm 0.13$ μm$^2$, vortex $0.715 \pm 0.120$ μm$^2$), maximum average intensities (middle, blip $874.0 \pm 96.4$, cloud $615.9 \pm 44.6$, vortex $1814.1 \pm 249.6$), and decay time constants (right, blip $5.16 \pm 0.41$ s, cloud $18.25 \pm 0.92$ s, vortex $18.50 \pm 2.33$ s) of the localized actin events (blip $n = 14$, cloud $n = 24$, vortex $n = 9$ from three biologically independent). **i** The number of occurrences of the three localized actin events (from 3 RAW264.7 cells). We used the two-tailed unpaired Student's $t$-test for data in (h). *$p < 0.05$, **$p < 0.01$, ***$p < 0.001$, ****$p < 0.0001$. $p = 0.0071$ (blip vs vortex), $p = 0.0198$ (cloud vs vortex) in the area; $p = 0.0448$ (blip vs cloud), $p = 0.0004$ (blip vs vortex), $p = 0.0009$ (cloud vs vortex) in the maximum average intensities; $p = 0.0005$ (blip vs cloud), $p = 0.0018$ (blip vs vortex) in the decay time constants.

live cells. Although these transient actin events were also observed by the SD-SIM that conferred lower resolution (Supplementary Fig. 14), we argued that improved resolution afforded by the BF-Sparse-SIM did lead to more resolvability. Indeed, while actin events observed by the SD-SIM showed maximal average intensities and decay kinetics similar to those yielded by BF-Sparse-SIM, we observed fewer vortex events with SD-SIM, and their sizes were larger. This is best explained by the lower resolution of SD-SIM compared to the BF-Sparse-SIM, which might only resolve large vortex structures and misclassify small ones as actin cloud events. On the other hand, people have found actin flashes associated with phagocytosis in macrophages[22]. However, compared to blips, clouds, and vortexes, they are much larger (~4 μm in diameter) and stay on the plasma membrane for extended periods (lifetime of

3–7 min)[22]. Alternatively, macrophages use actin tent poles followed by their cross-over to internalize fluid during micropinocytosis[23]. As actin tentpoles originated from linear ruffles at the cell periphery, they also differ from what we have observed here. Therefore, to our best knowledge, we argue that the actin blip, cloud, and vortex within the cortex of macrophages are three unreported structures.

As actin blips were similar in size and intensity to newly-added individual fluorescence punctum during the development of the actin cloud, the former may act as the precursor of the latter. Similar to the formation of clathrin-coated pits[44], random initiations of actin blips on the plasma membrane constitute the seed for the following reaction. Without neighboring polymerization sites activated, blips quickly disassemble like abortive pits[44]; upon encountering neighboring

polymerization sites, they are amplified into clouds that propagate laterally. Detailed analysis may assist in mathematically modeling this non-equilibrium process, such as how many activators are spatially placed around the initial seeding site to antagonize delayed inhibitions to catalyze the self-generating cloud[18]. Although actin polymerization by side branching was also observed in vortex events, it demonstrated a spiral and inward wavefront coupled with much-elevated peak fluorescence, suggesting a mechanism different from blips and clouds. It is also intriguing to observe two different polymerization speeds in the actin vortex; thus, actin's two distinct regulatory mechanisms and functions may operate collaboratively in one sequentially-occurring event. Nevertheless, quantitative assessment of setpoints before massive feedback amplification and disassembly of actin filaments will help determine critical parameters of the activator-inhibitor interacting model in time, which may provide insights into the complex, non-equilibrium actin dynamics.

Overall, BF-Sparse-SIM enables analysis of subcellular dynamics in live cells down to ~70 nm scale and provides insights. The whole BF-Sparse deconvolution pipeline can readily be adapted to other types of SR systems and fluorescence microscopes. For example, we also decomposed background fluorescence in widefield images as in-focal signals and out-of-focus background to improve contrast, and its conjunction with the sparse deconvolution substantially enhanced image resolution (Supplementary Fig. 17). Thus, it may be a generally helpful tool for future quantitative SR investigations of cell dynamics in live cells.

## Methods

### Cell maintenance and preparation

COS-7, U2OS, and RAW 264.7 cells were cultured in high-glucose DMEM (Gibco, 21063029) supplemented with 10% FBS (Gibco, VIS146975) and 1% 100 mM sodium pyruvate solution (Sigma, S8636) in an incubator at 37 °C with 5% $CO_2$ until ~75% confluency was reached. For the 2D-SIM imaging experiments, cells were seeded onto coverslips (H-LAF 10 L glass; reflection index, 1.788; diameter, 26 mm; thickness, 0.15 mm, customized). Detached using trypsin-EDTA (Gibco, 25300054), seeded onto poly-L-lysine (Sigma, P4707) coated coverslips, and then cultured for ~24 h before transfection. For the SD-SIM and STED imaging experiments, 25-mm coverslips (Fisherbrand, 12-545-102) were coated with 0.01% poly-l-lysine solution (Sigma, P4707) for ~24 h before seeding transfected cells. To label mitochondria, COS-7 cells were incubated with 250 nM MitoTracker Green FM (Thermo Fisher Scientific, M7514) or PK Mito Deep Red (Genvivo Biotech, PKMR-1) in HBSS solution containing $Ca^{2+}$ and $Mg^{2+}$ without phenol red (Thermo Fisher Scientific, 14025076) at 37 °C for 15 min before being washed and imaged. To label lipid droplets, COS-7 cells were incubated with 1 × LipidSpot 488 (Biotium, 70065-T) in a complete cell culture medium at 37 °C for 30 min, protected from light before being washed and imaged.

To label organelles with genetic indicators, COS-7 cells were transfected with plasmid LifeAct-EGFP and mCherry-Cytb5ER[45], and U2OS cells were transfected with LifeAct-EGFP. RAW264.7 cells were transfected with Lifeact-EGFP before conducting 2D-SIM imaging or SD-SIM imaging. COS-7 cells were transfected with GCaMP6s and seeded onto 25-mm coverslips (Citotest, 80344-2520). For calcium imaging experiments under the 2D-SIM configuration, calcium signal was stimulated with a micropipette containing 10 μM L$^{-1}$ 5′-ATP-Na$_2$ solutions (Sigma-Aldrich, A1852). In STED imaging experiments, cells were transfected with Lifeact-Halo and incubated with SiR for ~15 min without washing before imaging. The transfection was executed using Lipofectamine 2000 (Thermo Fisher Scientific, 11668019) according to the manufacturer's instructions. The transfected cells were cultured for 24 h, detached using trypsin-EDTA, seeded onto poly-l-lysine-coated coverslips, and cultured for an additional 18–24 h before imaging. Live cells were imaged in a complete cell culture medium without phenol red in a 37 °C live-cell imaging system.

Tetra Fluorescence Standard Sample slides (Standard Imaging, customization), BSC-1C cells were immunostained with β-tubulin E7 (GenFluor Green, Genvivotech) and used as biological test slides.

The source of cell lines used: COS-7 cells were kindly provided by Professor Heping Cheng, Peking University (ATCC, CRL-1651), U2OS cells were kindly provided by Professor Yu-Hui Zhang, Huazhong University of Science and Technology (ATCC, HTB-96), RAW264.7 cells were kindly provided by Professor Guangjun Nie, National Center for Nanoscience and Technology (ATCC, TIB-71), BSC-1C cells were kindly provided by Standard Imaging Company(ATCC, CCL-26).

### SIM imaging

A commercial structured illumination microscope (HIS-SIM, Guangzhou Computational Super-resolution Biotech) was used to acquire the cell images, which was based on a commercial inverted fluorescence microscope (IX83, Olympus), equipped with two objectives (×100/1.7 HI oil, APON, Olympus; ×100/1.49 oil, UAPON, Olympus), four excitation wavelength lasers (405 nm, 488 nm, 561 nm, and 640 nm), a multiband dichroic mirror (ZT405/488/561/640-phase R, Chroma) and an sCMOS camera (Flash 4.0 V3, Hamamatsu)[10]. HIS-SIM is controlled by its own software Imager (v1.1.23d). The raw image obtained has a pixel size of 65 nm. Generally, the exposure time of each frame of the raw image is 5 ms or 10 ms. The OTF used for reconstruction was prepared from actual fluorescent bead images (40 nm in diameter, ThermoFisher, F8771). And the diameter of the fluorescent microbeads used to compare the reconstruction effect of different exposure times, i.e., different signal-to-noise ratios, was 200 nm (ThermoFisher, F8811). In calcium imaging experiments, we collected the first 3–5 s as the basic fluorescence levels before adding ATP solutions to stimulate cells, and set the raw image exposure for 10 ms and imaging without interval for 2 min.

### The SD-SIM setup

The SD-SIM is a commercial system based on an inverted fluorescence microscope (IX81, Olympus) equipped with a widefield objective (×100/1.3 oil, Olympus) and a scanning confocal system (CSU-X1, Yokogawa). It used software MetaMorph (v7.8.1.0) to capture images. Four laser beams of 405 nm, 488 nm, 561 nm, and 647 nm were combined with the SD-SIM. The Live-SR module (GATACA systems) was equipped with the SD-SIM. The images were captured either by an sCMOS camera (C14440-20UP, Hamamatsu) or an EMCCD camera (iXon3 897, Andor).

### The STED setup

Stimulated Emission Depletion (STED) imaging was performed using a STEDYCON (Abberior Instruments) with excitation lasers at 450 nm, 594 nm, and 640 nm and a STED laser at 775 nm wavelength (all pulsed), and it was controlled by its own software, Smart Control (SN: SY210901). The STEDYCON was mounted at the camera port of a Zeiss Axio Observer Inverted microscope equipped with a ×100/1.40 oil, UPlanSApo objective. The excitation wavelength range was set to 640 nm for Lifeact-Halo. Depletion power was set to achieve a resolution of 40 nm, the pinhole was set at 64 μm, and the t-series was set to 100 frames with 5 s intervals. Data was stored in obf format and exported as tiff files for further analysis. All images were obtained using LAS AF software (Leica).

### Quantification of SR image contrasts and FRC resolution map

FRC is a method to calculate the effective resolution of an image by using the image properties that the signal satisfies the correlation and the noise satisfies the noncorrelation. To evaluate the resolution of the SR images and obtain the FRC resolution map, we used our recently developed ImageJ's plugin PANELJ, and set the FRC threshold mode to 3-sigma and block size to 128[40]. In addition, for unbiased cross-validation, we also used NanoJ-SQUIRREL to calculate the minimum and average FRC resolutions while setting the block size to the default

value of 10[41]. Details of SR image contrasts were given in Supplementary Note 1.

## Generation of 3D PSF

The simulation 3D-PSF used by BF-SIM was obtained by ImageJ's plugin PSF Generator. The relevant parameters were set as follows: refractive index immersion: 1.5, accuracy computation: good, wavelength: 488/561/640 nm, XY pixel size is the same as the SIM raw data: 65 nm, NA: 1.4 (used the actual effective imaging NA which was better than the objective lens nominal NA), Z-step: 100 nm, and the size X-Y-Z: 512-512-131. When generating the 3D PSF, we selected the Richards & Wolf 3D Optical model.

## Image processing and statistical analysis

Image reconstruction was primarily performed using MATLAB (MathWorks, 2014a), while fairSIM reconstruction was obtained by ImageJ, and DFCAN reconstruction was obtained by the trained F-actin model of DFCAN-SIM[39]. For the sparse deconvolution, we followed the procedure as elaborated in detail previously[11]. After loading data into the Sparse software, three parts of parameters need to be set, respectively: fixed parameters, image property parameters, and content-aware parameters. The fixed parameters and the image property parameters are selected based on the hardware system and the image property, such as high-or-low SNR and strong-or-weak background (Wavelets filtering), and need little tuning. The content-aware parameters mainly include image fidelity (the inverse of the $xy$ continuity, 1000–300 for high SNR images), sparsity (need to finetune this parameter back-and-forth), and iterative deconvolution times (5–15 for the RL algorithm and 30–50 for the LW algorithm), which need to be adjusted carefully to achieve the optimal reconstruction results[11]. We used ImageJ's plugin Threshold and AnalyzeSkeleton (2D/3D) to obtain the information on actin filaments. Upon threshold segmentation, we first converted the image into 8-bit data. Then we chose the Huang method or the automatic threshold method to binarize the image according to the actual image segmentation effect. We then used AnalyzeSkeleton (2D/3D) to analyze the skeletonized lengths of different actin filaments directly. The density of actin is the area of the segmented actin after binarization divided by the area of the entire cell distribution. As for the program STICS calculating the actin waves, it used the open-source software provided by Ashdown et al.[27], in which the parameters were set according to the actual imaging conditions. All data processing was achieved using ImageJ, such as the color map of images. All data were plotted, statistical tests, and final images were prepared using GraphPad Prism software (GraphPad Software, 8.3.0). Quantitative data are presented as box-and-whisker plots (centerline, average; limits, 75 and 25%; whiskers, maximum, and minimum), mean ± s.e.m. and scatter plot (median) graphs and tables. The statistical significance analysis of data was made by using the two-tailed student $t$-test method and no adjustment was made for multiple comparison, where ns $p \geq 0.05$ and * $p < 0.05$, ** $p < 0.01$, *** $p < 0.001$, **** $p < 0.0001$.

## Statistics and reproducibility

No statistical method was used to predetermine sample size. No data were excluded from the analyses. The experiments were not randomized. The Investigators were not blinded to allocation during experiments and outcome assessment. No less than three independent replicates were performed on the microscope images shown in the manuscript and in the supplementary information, such as Fig. 1b and supplementary Fig. 12a–f.

## Data availability

The raw images and reconstructed SR images from Figs. 1, 2e, f and Supplementary Figs. 3b, c, 6, 9, 10, and an example dataset from Figs. 3, 4 and Supplementary Fig. 12 data used in this study are available at https://doi.org/10.6084/m9.figshare.21792788. The source data of the figures (in the main manuscript and in the Supplementary Information) are provided as the Source Data files in this paper. Source data are provided in this paper.

## Code availability

The current version of the custom-written MATLAB code and its test example data used in this manuscript are available at https://doi.org/10.6084/m9.figshare.22640974.v1.

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

## Acknowledgements

This work was supported by the National Key Research and Development Program of China (2022YFC3400600 L.C.), the National Natural Science Foundation of China (T2288102 L.C., 81925022 L.C., 62103071 J.F., 32227802 L.C., 92150301 L.C., 92054301 L.C., 61827825 Y.Z., and 92054110 Y.Z.), the Beijing Natural Science Foundation Key Research Topics (Z20J00059 L.C.), the Natural Science Foundation of Chongqing (cstc2021jcyj-msxmX0526 J.F., sl202100000288 J.F.), Science and Technology Research Program of Chongqing Municipal Education Commission (Grant No. KJQN202100630 J.F.). We also thank the High-Performance Computing Platform of Peking University.

## Author contributions

Y.M. and J.F. carried out the experiments. Y.M. performed the data analysis and drafted the manuscript. K.W., L.L., S.X., S.Y., X.D., T.C., and Y.Z. prepared and conducted the cell experiments. J.W. helped with the sparse deconvolution of partial data. Z.L. helped with DFCAN reconstruction. W.G. helped with the Notch filtering SIM reconstruction. C.G., J.F., and L.C. conceived and designed the study and revised the manuscript. All authors read and approved the manuscript.

## Competing interests

S.Y., J.W., and Z.L. were employed by Guangzhou Computational Super-resolution Biotech Co., Ltd. The remaining authors declare no competing interests.
