## [Peer Review File · Nature Communications]

Quantitative structured illumination microscopy via a physical model-based background filtering algorithm reveals actin dynamicsREVIEWER COMMENTS

Reviewer #1 (Remarks to the Author):

Despite tremendous work on instrumental development and wonderful applications of structured illumination microscopy (SIM) in cell biology, its super-resolution (SR) image reconstruction algorithms remain a hot topic in recent studies. Quite a few papers were published for diminishing different types of artifacts. This paper due with the hammer-stroke and honeycomb artifacts caused by the out-of-focus signals. Before SIM reconstruction, out-of-focus components calculated from simulated 3D PSF were subtracted from raw SIM images, yielding a clean in-focus image. These in-focus images were used to perform SR reconstruction using previously developed algorithms, such as Hessian-SIM and Sparse-SIM. SR images with higher contrast and finer structures then could be obtained, as demonstrated with a few kinds of samples and comparison with several open-source SIM algorithms in the paper. The improved image quality helps the authors to observe the dynamic actin wave and discovered new types of localized actin dynamics in the cell cortex.

The work would be interesting to SIM developers and users in many fields. However, more detailed explanations and reasoning are needed to evidence the general validity of their algorithms.

(1) In the model, the authors assumed that the illumination patterns of the in-focus and out-of-focus in the (x-y) plane are the same. They also assumed that the sample structure doesn't change significantly in 3D. Without these two assumptions, the equation (7) in the Supplementary won't be correct. Could the authors give evidence or explanation on under what circumstance the two assumptions can be regarded as true? If not true (for example the 3D structure cannot be the same in 4 μm), to what extent the result will be affected?

(2) In the pseudo-code of BF-SIM, the first three steps just calculate of $H_{in}(k)$ and $H_{out}(k)$ from the simulated 3D PSF model. These are not related to the sample structure. In other words, for a given SIM system these $H_{in}(k)$ and $H_{out}(k)$ can be previously calculated before any experiment. Is that correct? Could the author also give out the calculated $H_{in}(k)$ and $H_{out}(k)$ from the 3D pdf model?

3) In BF-SIM, the author calculate the out-of-focus without taking a 3D stack imaging. Imagine a sample with a single layer of fluorescent beads fixed on the surface, there will be no out-of-focus signals in principle. But the same $H_{in}(k)$ and $H_{out}(k)$ will still be applied to this sample. In this circumstance, how will the final SR-image be affected with BF-SIM?

4) In Figure 3, the authors calculated the velocity of the actin main flow. To my mind, this is the first report of actin flow observed in a live cell, which is great. However, the authors need to describe how it is calculated in the Methods.

Reviewer #2 (Remarks to the Author):

In this article, Mo et al. developed a denoising algorithm that improves SIM reconstruction quality seemingly without losing the underlying signal linearity. Visually the results of the denoising process look very nice, and the approach here is likely to become very useful. However, I have several concerns/ comments. In its current form, it is very challenging for this reviewer to assess the quality of the work as 1) the material and methods section is mostly absent, and 2) the tool provided requires Matlab to run, which I do not have access to and therefore cannot test.

Main comments:

The authors should consider other methods to assess the quality of their reconstructions. For instance, NanoJ-SQUIRREL could help map more accurately possible artifacts introduced by their methods (or the lack of them) (<https://www.ncbi.nlm.nih.gov/pmc/articles/PMC5884429/>).

The authors show improved signal linearity using one dataset. I strongly suggest that the authors study the signal linearity for all their examples. Again, NanoJ-SQUIRREL could also help the authors further demonstrate better linearity.

The authors strongly emphasize that their method is "quantitative". This is problematic as, at best, the SIM reconstruction is linear(ish). The SIM reconstruction will still remove most of the non-structured signal. While some of it is due to noise, a large part will be from an unbound cytoplasmic protein pool (real biological relevant signal). This should be acknowledged in the text. This is especially problematic as the authors indicate in the discussion, "BF-SIM may be better suited for quantitative fluorescence imaging of cellular dynamics such as cytosolic Ca²⁺ concentrations and action potentials".

For the measurements of image resolution, it would be helpful to use unbiased-established methods such as FRC or decorrelation analysis (<https://www.nature.com/articles/s41592-019-0515-7>).

The authors claim that using their method provides a "More accurate and quantitative analysis". This is, however, not demonstrated here. Without a ground truth to validate their analysis, it is impossible to assess the accuracy of their analysis. SIM datasets are not always noisy; the authors should consider generating datasets containing simulated noise to demonstrate the validity of their approach. The authors could also consider acquiring the same sample using two illumination regimes (one with noise, one with low noise) to assess their reconstruction method. Alternatively, compare how their approach compares against non-linear SIM reconstruction algorithms.

The authors claim they discovered "novel, dynamic actin structures". This bold claim would deserve to be cross-validated using another super-resolution method.

It is disappointing that none of the test datasets are made publicly available. It would be essential to make these available to the community so that the results presented here could be replicated and used to develop SIM processing further.

In the discussion, it would be important to highlight the limitation of the current approach. For instance, can this denoising approach be used to denoise 3D SIM datasets? or only 2D SIM datasets with a plane-by-plane strategy?

Reviewer #3 (Remarks to the Author):

Mo and co-authors have devised a simple method to suppress out-of-focus background from raw 2D-SIM acquisitions, thereby achieving sectioning in a computational way. They demonstrate the usefulness of their computational sectioning method on several showcases, in particular on live actin, and even claim discovery in actin dynamics.

I think the basic idea is interesting and useful to the community, and will surely advance the potential of 2D SIM. I do, however, have several objections and questions that the authors should address.

1. Incomplete information. The assessment of what is background in the raw images is done with a Fourier space filtering technique. Unfortunately, the information on the procedure is very sketchy. It is necessary to show the actual in and out-of-focus transfer function $H_{in}(k)$ and $H_{out}(k)$ and also the filter function $H_{out}/(H_{in}+H_{out})$ appearing in eq. 3. It is also necessary to show examples of several raw images and how they split into the background and foreground components according to the proposal. Also, the impact of the criterion what is termed in or out of focus should be investigated in detail. Now this is an arbitrary user defined 400 nm limit, which is a major weakness of the method. I would like to see the impact of variations of this number on the above mentioned filters and decompositions, as well as on the final reconstruction results. How else are

we to know that we do not fall into the trap of "parameter tuning prone to human bias", which "fail at densely labeled fluorescent structures" that is the drawback of existing methods?

2. Lack of comparison to state-of-the-art. The authors do show how their method can benefit a variety of SIM reconstruction methods, which is highly appreciated. The technique of notch filtering is known to suppress out-of-focus background from 2D-SIM reconstructions [Wicker et al. , *Optics Express* 21, 2032–2049, 2013; . Muller et al., *Nature Communications* 7, 10980, 2016]. This is clearly the state-of-the-art method that is closest in spirit to the proposed idea. Yet, a direct comparison to this method is missing, even though this is the most relevant comparison to make. In what ways is the proposed method better? Can more reliable reconstructions be guaranteed with the proposed method? Can the authors show a side-by-side comparison of the best possible background suppression that can be achieved with either method? How sensitive would these cases be to parameter selection? In my opinion, such a detailed comparison is necessary.

3. Unsubstantiated claims.

a. The authors contrast their method to "selective frequency spectrum attenuation", but their background assessment is based on Fourier space filtering. Clearly, their method also qualifies as a "selective frequency spectrum attenuation" method, and should be named as such.

b. The sub-70 nm resolution claim is a bit too thin to my mind. It is based on a best result measured in a patch in one image. The authors should replace this claim with typical or average FRC values they find in multiple patches in multiple images. Also, it is known that non-linear processing techniques, such as deconvolution, can result in a better FRC resolution (higher precision), but can come at the expense of a loss in accuracy, as structures are presented in the reconstruction that do not represent reality (see e.g. ref 14). This makes the low sub-70 nm value perhaps not so valuable. A propos, one or more references to original work on FRC in electron and/or optical microscopy would also be appropriate.

c. The key showcase on live actin (Fig. 4) is analyzed using a sparse deconvolution technique proposed by the same group (ref. 11). Unfortunately, this confounds their conclusions. Can the claimed effects on actin dynamics be seen owing to the sparse deconvolution technique, to the proposed background suppression method, or is a combination of the two essential? The actual images shown with the sparse SIM technique do not seem artefact free to me. Ways to tackle these questions is to give FRC numbers with/without the background suppression method, and to do the same analysis on the live actin dataset by combining the method with standard SIM reconstruction, so as to eliminate deconvolution artefacts. If their conclusions on actin blips etc. then still stand it would make their cases stronger.

4. I would also like to note that I disagree with the data availability statement "upon reasonable request". I prefer fully open data, or a statement without this questionable disclaimer of "reasonable".

Finally, I cannot assess the credibility of the novelty claims made on actin dynamics, as I am no expert in actin biophysics.

RESPONSE TO REVIEWERS' COMMENTS

We thank all the reviewers for their comments regarding our manuscript (NCOMMS-22-31115A). As a background-suppressed method for SIM, we understand your concerns regarding its underlying principles, superiority over other methods, and limitations. Therefore, with the help of colleagues, we confronted these doubts and conducted challenging experiments to address each of them. In the updated manuscript, we have added **8 new Supplementary Figures (1, 2, 3, 5, 6, 13, 14, 15, 16)** and **2 new Supplementary Tables (1, 2)**. We have also revised **Supplementary Note 1 and Supplementary Figure 11** and included more discussion as suggested by the reviewers. The major revisions to the paper are listed below:

1) We have explained the principle underlying the BF model and showed its superiority over other background suppression methods in retaining weak signals and removing background fluorescence. We have provided the entire deduction of the BF-SIM model, including the consistency of the z-axis illumination of 2D-SIM and the approximation between the in- and out-of-focus signals after convolution with the defocused PSF (**Supplementary Note 1, Supplementary Fig. 1, 2**). We showed that an in-focal range of $z_{in} = \pm 0.4 \mu\text{m}$ is generally applicable for handling images of different organelles (**Supplementary Fig. 3**). Although we proposed the BF method for widefield and two-beam interference SIM (**Supplementary note 1**), its application in low-background images, such as actin filaments under the GI-SIM or TIRF-SIM (with the axial thickness no more than $1 \mu\text{m}$, **Response Fig. 1**), fluorescence beads on the slide surface (**Supplementary Fig. 5**), and obtained by three-beam interference 3D-SIM (**Response Fig. 2**) still retained weak signals while suppressing artifacts and enhancing contrast.

2) From multiple perspectives, we have established that the BF-SIM is a quantitative SR method. Using NanoJ-SQUIRREL to calculate RSE as a linearity measurement, our BF-SIM showed good RSE for actin filaments and fluorescent beads (**Supplementary Fig. 5, 6**), which surpassed other methods that needed fine-tuning. Moreover, we compared the linearities of different lipid droplets (**Fig. 1g and 1h**) and cytosolic Ca^{2+} signals (**Supplementary Fig. 16**) by various reconstruction methods, highlighting that BF-SIM achieved linearity levels similar to the original Wiener-SIM. Finally, we analyzed the kinetics of actin blips, clouds, and vortexes in RAW264.7 cells obtained by the BF-Sparse-SIM, which were similar to those obtained by the Wiener-SIM (**Supplementary Fig. 13**). These data reinforced the validity of BF-Sparse-SIM in reporting quantitatively accurate kinetics.

3) Per the Reviewers' comments, we have performed new experiments to validate the novel actin dynamics observed in RAW264.7 cells by the BF-Sparse-SIM. We could still distinguish the actin blip, cloud, and vortex examples under the conventional Wiener-SIM, which yielded kinetics similar to those from BF-Sparse-SIM reconstructions (**Response Fig. 3, Supplementary Table 1**). Moreover, we also used the SD-SIM to detect actin blip, cloud, and vortex events resembling those observed by BF-Sparse-SIM (**Supplementary Fig. 14, Supplementary Table 2**).

In addition to these modifications, we have included more materials and methods in the manuscript and made datasets in the paper publicly available. In the Point-to-point responses, we have also assembled 3 more **Response Figures** and other materials to address all previous concerns. With these

revisions, we hope to convince you that we have provided sufficient evidence to demonstrate the usefulness, superiority, and linearity of the BF-SIM and BF-Sparse-SIM. Thus we hope you will agree to publish it in *Nature Communications* so people can readily explore its potential in their studies.

A list of newly added Figures

Supplementary Figure 1: The schematic illustration of two-beam illumination for 2D-SIM.

Supplementary Figure 2: In-focal widefield distributions of 3D samples and the defocus background calculated from different image depths.

Supplementary Figure 3: Different \mathbf{H}_{in} , \mathbf{H}_{out} and $\mathbf{H}_{out}/(\mathbf{H}_{in} + \mathbf{H}_{out})$ corresponding to different in-focus depths of BF-SIM.

Supplementary Figure 5: Fluorescence beads of various SNRs reconstructed with different background suppression methods.

Supplementary Figure 6: Back-to-back comparisons among BF-SIM, HiFi-SIM, and NF-SIM.

Supplementary Figure 13: Resolutions and actin event kinetics obtained by BF-Sparse-SIM, Wiener-SIM, and BF-SIM.

Supplementary Figure 14: Actin blips, clouds, and vortexes observed by the SD-SIM.

Supplementary Figure 15: Actin clouds observed by the STED.

Supplementary Figure 16: Ca^{2+} transients under the BF-SIM are highly correlated to those under the widefield microscopy.

Supplementary Table 1: Actin dynamics revealed by BF-Sparse-SIM, Wiener-SIM, and BF-SIM.

Supplementary Table 2: Actin dynamics revealed by BF-Sparse-SIM, SD-SIM, and STED.

Response Figure 1: The BF algorithm helps suppress artifacts in nonlinear GI-SIM reconstructions.

Response Figure 2: We applied the BF method in handling the 3D-SIM dataset.

Response Figure 3: A representative example of an actin vortex event that increased fluorescence intensities followed the spiral and outward-to-inward pattern.

REVIEWER COMMENTS

Reviewer #1 (Remarks to the Author):

Despite tremendous work on instrumental development and wonderful applications of structured illumination microscopy (SIM) in cell biology, its super-resolution (SR) image reconstruction algorithms remain a hot topic in recent studies. Quite a few papers were published for diminishing different types of artifacts. This paper due with the hammer-stroke and honeycomb artifacts caused by the out-of-focus signals. Before SIM reconstruction, out-of-focus components calculated from simulated 3D PSF were subtracted from raw SIM images, yielding a clean in-focus image. These in-focus images were used to perform SR reconstruction using previously developed algorithms, such as Hessian-SIM and Sparse-SIM. SR images with higher contrast and finner structures then could be obtained, as demonstrated with a few kinds of samples and comparison with several open-source SIM algorithms in the paper. The improved image quality helps the authors to observe the dynamic actin wave and discovered new types of localized actin dynamics in the cell cortex.

The work would be interesting to SIM developers and users in many fields. However, more detailed explanations and reasoning are needed to evidence the general validity of their algorithms.

Reply 1: Thank you for the appraisal of "improved image quality helps the authors to observe the dynamic actin wave and discovered new types of localized actin dynamics in the cell cortex" and your regards of the work as "would be interesting to SIM developers and users in many fields." In the updated manuscript, we have updated more explanations and validation of the general applicability of our algorithm, as you and the other reviewers have suggested. We hope to convince you that we have adequately addressed all your concerns.

(1) In the model, the authors assumed that the illumination patterns of the in-focus and out-of-focus in the (x-y) plane are the same. They also assumed that the sample structure doesn't change significantly in 3D. Without these two assumptions, the equation (7) in the Supplementary won't be correct. Could the authors give evidence or explanation on under what circumstance the two assumptions can be regarded as true? If not true (for example the 3D structure cannot be the same in 4 μm), to what extent the result will be affected?

Reply 2: Thank you for the reminder, and we apologize that we did not clearly describe the underlying assumptions in the original manuscript. Now we have added discussion as below:

1) For illumination pattern of 2D-SIM

*2D-SIM uses two coherent beams with the same polarization direction (such as S-polarized light) for illumination, as shown in **Supplementary Fig. 1**. Assuming that the wavevectors of the two beams are \vec{k}_1 and \vec{k}_2 , respectively, the two beams can be expressed as*

$$\begin{aligned}\vec{E}_1 &= E_1 \exp[i(\vec{k}_1 \cdot \vec{R} - \omega t)], \\ \vec{E}_2 &= E_2 \exp[i(\vec{k}_2 \cdot \vec{R} - \omega t + \varphi)],\end{aligned}$$

where E_1 and E_2 are the amplitudes of two coherent beams, ω is the angular frequency, and φ is the phase difference between the two beams that produces illumination patterns with different phases. And $\vec{R} = (x, y, z)$ is the space position vector. Thus, the illumination intensity I could be written as:

$$I = |\vec{E}_1 + \vec{E}_2|^2 = |\vec{E}_1|^2 + |\vec{E}_2|^2 + 2|\vec{E}_1 \cdot \vec{E}_2| = I_1 + I_2 + 2\sqrt{I_1 I_2} \cos[(\vec{k}_1 - \vec{k}_2) \cdot \vec{R} + \varphi],$$

where I_1 and I_2 are illumination intensities of two beams. According to the properties of coherent light $|\vec{k}_1| = |\vec{k}_2| = k_0$, $I_1 = I_2 = I_0$. Therefore, the illumination intensity after interference could be represented as follows:

$$I = 2I_0 + 2I_0 \cos[2k_0 \sin\theta \cdot x + 2k_0 \sin\theta \cdot y + \varphi].$$

As seen from the above equation, the illumination intensity distribution is independent of the z-axis; in other words, the illumination at different depths of the z-axis of 2D-SIM is the same when we do not consider the illumination aberration. Therefore, under the 2D-SIM illumination, the BF-SIM model is consistent with the actual situation.

2) Approximately invariant intensities of the sample in the Z-axis

The image taken by the camera is equal to the accumulation of objects of different depths convolved with 3D-PSF of corresponding depths:

$$d(\mathbf{r}) = [g(\mathbf{r}, z_{in}) \cdot I(\mathbf{r})] \otimes h(\mathbf{r}, z_{in}) + [g(\mathbf{r}, z_{out}) \cdot I(\mathbf{r})] \otimes h(\mathbf{r}, z_{out}). \quad (P1)$$

Here we do not assume that the sample structure stays the same in 3D ($g_{out} \approx g_{in}$). Instead, we make an integral approximation that:

$$[g(\mathbf{r}, z_{out}) \cdot I(\mathbf{r})] \otimes h(\mathbf{r}, z_{out}) \approx [g(\mathbf{r}, z_{in}) \cdot I(\mathbf{r})] \otimes h(\mathbf{r}, z_{out}). \quad (P2)$$

Therefore, samples do not need to have the same distribution at different depths but would have approximately the same distribution after the convolution with h_{out} . We used multilayer 2D-SIM to image 3D cell samples with different distributions at different depths. Similar to the manuscript, we took ± 400 nm in the middle layer as in-focus signals and other layers as the out-of-focus background. We calculated the similarity between $[g(\mathbf{r}, z_{out}) \cdot I(\mathbf{r})] \otimes h(\mathbf{r}, z_{out})$ and $[g(\mathbf{r}, z_{in}) \cdot I(\mathbf{r})] \otimes h(\mathbf{r}, z_{out})$ (**Supplementary Fig. 2**). As can be seen from the statistical results, the hypothesis $[g(\mathbf{r}, z_{out}) \cdot I(\mathbf{r})] \otimes h(\mathbf{r}, z_{out}) \approx [g(\mathbf{r}, z_{in}) \cdot I(\mathbf{r})] \otimes h(\mathbf{r}, z_{out})$ is approximately satisfied.

Finally, we tested whether it was possible to estimate out-of-focal fluorescence intensity with the image from a single 2D layer. Thus, we denoted a single-layer 2D-SIM image as $g(\mathbf{r}, z_{focus})$, and calculate $[g(\mathbf{r}, z_{focus}) \cdot I(\mathbf{r})] \otimes h(\mathbf{r}, z_{out})$. As shown in **Supplementary Fig. 2c-2f**, $[g(\mathbf{r}, z_{focus}) \cdot I(\mathbf{r})] \otimes h(\mathbf{r}, z_{out})$ was almost the same as $[g(\mathbf{r}, z_{in}) \cdot I(\mathbf{r})] \otimes h(\mathbf{r}, z_{out})$. Thus, we have the following approximate relationship:

$$\begin{aligned} [g(\mathbf{r}, z_{focus}) \cdot I(\mathbf{r})] \otimes h(\mathbf{r}, z_{out}) &\approx [g(\mathbf{r}, z_{in}) \cdot I(\mathbf{r})] \otimes h(\mathbf{r}, z_{out}) \\ &\approx [g(\mathbf{r}, z_{out}) \cdot I(\mathbf{r})] \otimes h(\mathbf{r}, z_{out}). \end{aligned} \quad (P3)$$

Therefore, the final imaging model of Eq.(P1) can be rewritten as follows:

$$d(\mathbf{r}) = [g(\mathbf{r}, z_{focus}) \cdot I(\mathbf{r})] \otimes h(\mathbf{r}, z_{in}) + [g(\mathbf{r}, z_{focus}) \cdot I(\mathbf{r})] \otimes h(\mathbf{r}, z_{out}). \quad (P4)$$

Then, we may estimate the out-focal fluorescence from the in-focal single-plane image.

We have incorporated parts of the discussion above in the **Supplementary Note 1 (Page 2, Lines 12-29, Page 3, Lines 17-26, Page 4, Lines 1-8 and Lines 16-30)**.

3) For 3D-SIM:

Under the 3D-SIM with three-beam illumination, the illumination on the z-axis has different intensities at different depths, which does not meet our assumptions. In addition, 3D-SIM uses 3D PSF to perform 3D deconvolution of images during reconstruction, which essentially redistributes the background to its focal plane. Therefore, we do not think BF-SIM is valid or required for background suppression in 3D-SIM.

We have incorporated parts of the discussion above in the **Supplementary Note 1 (Page 5, Lines 2-10)**.

(2) In the pseudo-code of BF-SIM, the first three steps just calculate of $H_{in}(\mathbf{k})$ and $H_{out}(\mathbf{k})$ from the simulated 3D PSF model. These are not related to the sample structure. In other words, for a given SIM system these $H_{in}(\mathbf{k})$ and $H_{out}(\mathbf{k})$ can be previously calculated before any experiment. Is that correct? Could the author also give out the calculated $H_{in}(\mathbf{k})$ and $H_{out}(\mathbf{k})$ from the 3D pdf model?

Reply 3: Firstly, independent of the imaging sample, the 3D PSF of the microscope system is a stable distribution after the objective lens:

$$h(x, y, z) = \left| C \int_0^1 J_0 \left[k \frac{NA}{n_i} (x^2 + y^2)^{0.5} \rho \right] e^{-\frac{1}{2}jk\rho^2 z \left(\frac{NA}{n_i}\right)^2} \rho d\rho \right|^2,$$

where C is a normalizing constant, J_0 is the Bessel function of the first kind of order zero, k is the wavenumber in the vacuum of the emitted light, NA is the numerical aperture of the objective lens, n_i is the refractive index of the immersion layer¹. Based on this equation, we used ImageJ's plugin PSF Generator to generate our 3D PSF². Therefore, we can easily pre-calculate $H_{in}(k)$ and $H_{out}(k)$. In other words, in the same microscope system, different samples are convolved with the same PSF to obtain the corresponding raw image.

Secondly, per your suggestion, we have shown the actual in-focus and out-of-focus optical transfer function $H_{in}(k)$, $H_{out}(k)$, and the filtering function $H_{out}(k)/(H_{in}(k) + H_{out}(k))$, respectively (**Supplementary Fig. 3**). By calculating distributions of out-of-focus background and in-focus signals with different $H_{in}(k)$ and $H_{out}(k)$, we conclude that $z_{in} = \pm 0.4\mu\text{m}$ is the ideal choice with a relatively small RSE, better background suppression, and good contrast (**Supplementary Fig. 3**). Therefore, except for the nonlinear GI-SIM and TIRF-SIM dataset (**Response Fig. 1**), we set the parameter to $\pm 0.4\mu\text{m}$ throughout the paper, including new experiments done during this round of revision.

(3) In BF-SIM, the author calculated the out-of-focus without taking a 3D stack imaging. Imagine a sample with a single layer of fluorescent beads fixed on the surface, there will be no out-of-focus signals in principle. But the same $H_{in}(k)$ and $H_{out}(k)$ will still be applied to this sample. In this circumstance, how will the final SR-image be affected with BF-SIM?

Reply 4: Thanks for your suggestion. As you said, our BF-SIM is a method for calculating the defocus background; if the imaging background is negligible, such as under the illumination of TIRF-SIM, it does not need BF processing. However, in actual experiments, we cannot always guarantee little background in the TIRF-SIM images, no matter whether they are fluorescent beads or cells. Therefore, users may try our method first without considering the model's applicability.

Per your suggestion, we have also experimented with beads with little defocusing background, which was theoretically inapplicable. We spread beads on the slide surface, captured high and low SNR images with different exposure times, and then reconstructed them with different methods. Under standard contrast conditions, the reconstruction was acceptable for Wiener-SIM. However, if we tried either high or low SNR images of high contrast, artifacts manifested in the reconstructions due to the imperfect matching of the OTF used for reconstruction and the aberrated system OTF (**Supplementary Fig. 5**). Using the notch filtering SIM (NF-SIM, $g(k) = 1 - \text{attstrength} \cdot \exp[-|k|^2/(2 \cdot \text{attwidth}^2)]$) reduced some artifacts. However, residue artifacts persisted even if we set parameters to the optimal $\text{attStrength}=0.999$ and $\text{attwidth}=2$ as suggested in the original paper³. In addition, it was challenging for NF-SIM to achieve a good compromise between removing the background and retaining the signal in processing complex structures (**Supplementary Fig. 6**).

In contrast, our BF-SIM with the same parameter as in the previous manuscript eliminated artifacts, as HiFi-SIM did with the best parameter ($\text{attStrength}=0.92$). We used NanoJ-SQUIRREL⁴ to calculate the signal error RSEs between different reconstructions and widefield images, and the correlation coefficients (R squared of linear fitting) between the incremental fluorescence intensity ΔF_{SIM} of SR reconstruction and the incremental $\Delta F_{widefield}$. While there were no difference in R^2 among different reconstructions, the RSE of BF-SIM was higher than those of Wiener-SIM and NF-SIM, and was not significantly different from that of HiFi-SIM. Therefore, even if the model may not apply to the background-free experiments, BF-SIM can still retain the signal while achieving an RSE and R^2 similar to the previous fine-tuned state-of-the-art method.

(4) In Figure 3, the authors calculated the velocity of the actin main flow. To my mind, this is the first report of actin flow observed in a live cell, which is great. However, the authors need to describe how it is calculated in the Methods.

Reply 5: Thank you. The STICS (Spatio-temporal Image Correlation Spectroscopy) is a previously published open-source software that quantitatively assesses the F-actin flow in living T cells of SIM⁵⁻⁷. The STICS uses the spatiotemporal correlation spectroscopy characteristics of fluorescence signal to calculate the velocity and direction of actin flowing. It first quantifies the signal intensity fluctuation δi in the image by using the difference of pixel intensity in time and space relative to the mean value:

$$\delta i(x, y, t) = i(x, y, t) - [i],$$

where δi is the difference between pixel intensity at a single point in space and time versus the mean intensity, $[i]$ is the average value. The variables x , y , t represent the interest region in x and y directions, and time t , respectively.

And then define a correlation function for the interest region:

$$G_{STICS}(\zeta, \eta, \tau) = (\delta i[x, y, t] \delta i[x + \zeta, y + \eta, t + \tau]_{xy}) / ([i]_t [i]_{t+\tau}),$$

where ζ and η represent the space shifts in the x , and y directions, respectively, and τ is the time lag. Denominator square brackets represent the mean intensities. The peak position of the correlation function in this correlative region can provide information about signal flow and population diffusion, and the valuation of the peak value of the correlation function describes the speed of signal flow. Therefore, the successful extraction of actin velocity by the STICS algorithm critically depends on correctly-calculated fluorescence intensities of actin filaments, which is where BF-SIM reconstruction excels.

For the specific STICS calculation process and the points to note when using STICS to analyze the living cell data of SIM, please refer to references⁵⁻⁷. We have added the STICS method to the **Materials and Methods** in the manuscript (Page 11, Lines 29-31).

Reviewer #2 (Remarks to the Author):

1. In this article, Mo et al. developed a denoising algorithm that improves SIM reconstruction quality seemingly without losing the underlying signal linearity. Visually the results of the denoising process look very nice, and the approach here is likely to become very useful. However, I have several concerns/ comments. In its current form, it is very challenging for this reviewer to assess the quality of the work as 1) the material and methods section is mostly absent, and 2) the tool provided requires Matlab to run, which I do not have access to and therefore cannot test.

Reply 1: Thank you for the comment, "Visually the results of the denoising process look very nice, and the approach here is likely to become very useful." We agree with you that we shall make more efforts to make the algorithm more easily accessible for other investigators. Therefore, per your suggestions, we have added more detailed procedures and protocols in the revised materials and supplementary materials. To avoid installing MATLAB, we have also created a directly executable file compiled from MATLAB code, which is convenient for readers to use and test. The code and its corresponding test data samples are open-sourced and can be downloaded from <https://doi.org/10.6084/m9.figshare.21792824> and <https://doi.org/10.6084/m9.figshare.21792854>. However, you may need to download the cost-free MATLAB Component Runtime (MCR, R2018b with version 9.5) from the website "<https://ww2.mathworks.cn/products/compiler/matlab-runtime.html>" and install the MCR to run our executable file. We have also included detailed guidelines in the readme file packed in the code package in supplementary materials. This paper's first author is unfamiliar with Java, but we would very much like to help if someone wants to convert the code into the ImageJ plugin or integrate it into the fairSIM in the future.

We have incorporated parts of the materials above in the **manuscript (Page 9, Lines 19-25 and Lines 30-32, Page 10, Lines 1-3 and Lines 12-32, Page 11, Lines 1-18 and Lines 25-31, Page 12, Lines 7-12).**

Main comments:

2. The authors should consider other methods to assess the quality of their reconstructions. For instance, NanoJ-SQUIRREL could help map more accurately possible artifacts introduced by their methods (or the lack of them) (<https://www.ncbi.nlm.nih.gov/pmc/articles/PMC5884429/>).

Reply 2: Thanks for your suggestion. Now we have used NanoJ-SQUIRREL to reveal the error maps (resolution scale error, RSE) between the reconstructed SR image and widefield image. We have also used NanoJ-SQUIRREL to re-evaluate the FRC resolutions of reconstructed images. We have included updated RSE results in **Supplementary Fig. 3, 5, 6, and FRC resolutions in Supplementary Fig. 11, 13.**

3. The authors show improved signal linearity using one dataset. I strongly suggest that the authors study the signal linearity for all their examples. Again, NanoJ-SQUIRREL could also help the authors further demonstrate better linearity.

Reply 3: Point taken. Now we realize that the RSE calculated by NanoJ-SQUIRREL corresponds to the linear relationship between the pseudo widefield images and the reconstructed SR images⁴. Therefore, we have shown that our BF-SIM showed good RSE for actin filaments, mitochondria, and fluorescent beads (**Supplementary Fig. 3, 5, 6, detailed in Reply 6**). In addition, we showed that BF-SIM yielded an RSE for the actin image only smaller than the notch-filtering with an $\text{attStrength}=0.999$ (NF-SIM 0.999), while the latter removed weak filaments and was not applicable (**Supplementary Fig. 6**). Moreover, we have also compared the linearities of lipid droplets of different fluorescent intensities (**Fig. 1g and 1h**) and cytosolic Ca^{2+} signals of different amplitudes under various

reconstructions to the widefield images (**Supplementary Fig. 16**, see also **Reply 4**). All these data pointed out that BF-SIM achieved linearity levels similar to the Wiener-SIM, while HiFi-SIM performed the worst. Finally, per **Reviewer 3's** suggestion, we also calculated the kinetics of actin blip, cloud, and vortex events in RAW264.7 cells under different reconstructions. Decay time constants and the maximum intensities of three types of actin events under the Wiener-SIM showed similar trends to those obtained by the BF-Sparse-SIM (**Supplementary Fig. 13** versus **Fig. 4h**). Although the maximum intensity obtained by the Wiener-SIM were the most prominent (possibly due to the inclusion of integrated background fluorescence), three methods yield almost identical decay time constants (**Supplementary Fig. 13d**). These data reinforces the validity of BF-Sparse-SIM in reporting quantitatively accurate kinetics.

4. The authors strongly emphasize that their method is "quantitative". This is problematic as, at best, the SIM reconstruction is linear(ish). The SIM reconstruction will still remove most of the non-structured signal. While some of it is due to noise, a large part will be from an unbound cytoplasmic protein pool (real biological relevant signal). This should be acknowledged in the text. This is especially problematic as the authors indicate in the discussion, "BF-SIM may be better suited for quantitative fluorescence imaging of cellular dynamics such as cytosolic Ca²⁺ concentrations and action potentials".

Reply 4: Thank you. We agree that the BF-SIM algorithm may remove some signals originating from the unbound cytoplasmic protein pool, as we have stated in the text in the current **manuscript (Page 7, Lines 30-31)**. As we have explained in **Reply 3**, we have shown the BF-SIM algorithm provided SR reconstruction of different structures with linearity levels similar to the Wiener-SIM, better than other background-suppressing algorithms such as HiFi-SIM and NF-SIM. These data agreed with the actin velocities measured by the STICS algorithm, the accuracy of which depended on the fluctuations of actin fluorescence intensities. Finally, we conducted 2D-SIM imaging of cytosolic Ca²⁺ signaling in live COS-7 cells⁸ (**Supplementary Fig. 16**), confirming the BF-SIM algorithm's superiority over others. Therefore, we feel that it is justified to conclude that "Compared to other artifact-suppressing algorithms, BF-SIM may be better suited for quantitative fluorescence imaging of cellular dynamics such as cytosolic Ca²⁺ concentrations and action potentials".

We have incorporated parts of the materials above in the **manuscript (Page 7, Lines 31-32, Page 8, Lines 1-4)**.

5. For the measurements of image resolution, it would be helpful to use unbiased-established methods such as FRC or decorrelation analysis (<https://www.nature.com/articles/s41592-019-0515-7>).

Reply 5: Thank you. In our early version, we measured the FWHMs of actin filaments and calculated the minimum FRC value to represent the resolution of the SR images using a self-written PanelJ plugin, which is released in Biorxiv⁹ and is also under review in Nature Communications. Per your suggestion, we used NanoJ-SQUIRREL to re-analyze the images' minimum and mean FRC values (**Supplementary Fig. 11e, 13a**), confirming the results obtained with PanelJ (**Supplementary Fig. 11d**).

6. The authors claim that using their method provides a "More accurate and quantitative analysis". This is, however, not demonstrated here. Without a ground truth to validate their analysis, it is impossible to assess the accuracy of their analysis. SIM datasets are not always noisy; the authors should consider generating datasets containing simulated noise to demonstrate the validity of their approach. The authors could also consider acquiring the same sample using two illumination regimes (one with noise, one with low noise) to assess their reconstruction method.

Alternatively, compare how their approach compares against nonlinear SIM reconstruction algorithms.

Reply 6: Thanks for your suggestion. We proposed the BF-SIM to handle the artifacts caused by the defocusing background but not those caused by random noise, which could be suppressed by the Hessian spatiotemporal continuity penalty. Nevertheless, per your suggestion, we have also conducted experiments on fluorescent beads with long or short exposure times to test the performance of our algorithm under different noise levels. While there were no differences in R^2 (R squared of linear fitting) among different reconstructions, the RSE of BF-SIM was higher than those of Wiener-SIM and NF-SIM, and was not significantly different from that of HiFi-SIM (**Supplementary Fig. 5**, detailed in **Reply 9**). Therefore, even if the model may not apply to low-background situations, BF-SIM can still retain the signal while achieving an RSE and R^2 similar to the previous fine-tuned state-of-the-art method.

Per your suggestion, we also tested nonlinear SIM images from the BioSR dataset, captured using GI-SIM or TIRF-SIM^{10,11}. Because the thickness of these images was limited ($0.1-1 \mu\text{m}$), we needed to adjust the focal range $z_{in} = \pm 0.2 \mu\text{m}$ and the out-of-focus range $z_{out} = \pm 0.5 \mu\text{m}$. We used the open source code by Gang Wen et al¹² for the nonlinear Wiener reconstruction. Compared with Wiener-NLSIM, there were no prominent high-frequency bright spots in the spectrum domain (**Response Fig. 1**). As a result, fewer honeycomb artifacts manifested in the reconstructed SR image (red arrows), and the contrast improved as well. Therefore, BF background suppression could also improve the image fidelity of nonlinear 2D-SIM.

Response Fig. 1 | The BF algorithm helps suppress artifacts in nonlinear GI-SIM reconstructions. a, The widefield image of actin filaments and corresponding SR images reconstructed by Wiener-SIM without or with BF preprocessing. **b**, Corresponding reconstructed spectrums. SR reconstruction extended the frequency spectrum 3 times more than the widefield. However, significant high-frequency bright spots (shown by the yellow arrow) remained in the Wiener-NLSIM spectrum, which could be effectively suppressed with BF. **c**, Resolutions of widefield, Wiener-NLSIM, and BF-NLSIM calculated with NanoJ-SQUIRREL. **d**, Normalized contrasts of Wiener-NLSIM and BF-NLSIM. *, $p < 0.05$, according to the paired student t -test. **e**, The enlarged region in (a), in which Wiener-NLSIM

showed artifacts (indicated by the red arrow).

7. The authors claim they discovered "novel, dynamic actin structures". This bold claim would deserve to be cross-validated using another super-resolution method.

Reply 7: Thank you. Per your suggestions, we have imaged the related actin dynamic phenomenon with SD-SIM and STED, two other SR microscopy techniques that can perform live cell imaging. Under the SD-SIM, we also observed actin blips, clouds, and vortexes in RAW264.7 cells, with maximal average intensities and decay kinetics similar to those yielded by BF-Sparse-SIM (**Supplementary Fig. 14**). However, we observed fewer vortex events with SD-SIM, and their sizes were larger. This is probably due to the lower resolution of SD-SIM compared to the BF-Sparse-SIM, which might only resolve large vortex structures and misclassify small ones as actin cloud events.

Under the STED, however, we rarely observed actin vortexes or blips. Only large actin cloud events could be appreciated (**Supplementary Fig. 15**), which might be due to the more severe photo-toxic nature of the STED imaging condition. Nevertheless, we have assembled parameters of actin dynamics measured by different imaging methods, as shown in **Supplementary Table 2**. These data supported the validity of the novel actin structures we found and highlighted the advantage of BF-Sparse-SIM in accurately reporting these dynamic structures.

We have incorporated parts of the discussion above in the **manuscript (Page 7, Lines 16-21, Page 8, Lines 13-18)**.

8. It is disappointing that none of the test datasets are made publicly available. It would be essential to make these available to the community so that the results presented here could be replicated and used to develop SIM processing further.

Reply 8: Agreed. We have uploaded raw images and reconstructed SR images from **Fig. 1, 2e, f, and Supplementary Fig. 3b,c, 6, 9, 10** at <https://doi.org/10.6084/m9.figshare.21792788>. In addition, we have also provided individual example datasets from **Fig. 3, 4, and Supplementary Fig. 12** there.

9. In the discussion, it would be important to highlight the limitation of the current approach. For instance, can this denoising approach be used to denoise 3D SIM datasets? or only 2D SIM datasets with a plane-by-plane strategy?

Reply 9: Thank you. We designed the BF algorithm to suppress the background in the two-beam illumination for 2D-SIM and widefield microscopy. 3D-SIM uses three-beam illumination with a modulation pattern in the z-axis, which does not meet the assumption that the illumination distribution in the z-axis is the same in our model. 3D-SIM uses a 3D PSF to perform 3D deconvolution of images during reconstruction, essentially a signal redistribution process that restores the background to its signal depth. Therefore, it is unnecessary to do extra background suppression for 3D-SIM. The same shall apply to GI-SIM, TIRF-SIM, or any fluorescence images without significant background.

However, the BF algorithm might be used to suppress artifacts manifested in reconstructions under GI-SIM or TIRF-SIM (**Response Fig. 1, Reply 6**). In addition, even if we tried our method in a 3D-SIM dataset for a plane-by-plane background removal and reconstruction strategy, we did not see any signal loss due to model mismatch (**Response Fig. 2**). Finally, per your and **Reviewer 1's** suggestion, we have experimented with beads with little defocusing background, which was theoretically inapplicable. We spread beads on the slide surface, captured high and low SNR images with different exposure times, and reconstructed them with different methods. Under standard contrast conditions, the reconstruction was acceptable for Wiener-SIM. However, if we tried either high or low SNR images of high contrast, artifacts manifested in the reconstructions due to the imperfect matching of the OTF used

for reconstruction and the aberrated system OTF (**Supplementary Fig. 5**). Using the notch filtering SIM (NF-SIM, $g(k) = 1 - \text{attstrength} \cdot \exp[-|k|^2 / (2 \cdot \text{attwidth}^2)]$) reduced some artifacts. However, residue artifacts persisted even if we set parameters to the optimal $\text{attStrength}=0.999$ and $\text{attwidth}=2$ as suggested in the original paper³. In contrast, our BF-SIM with the same parameter used previously in the manuscript eliminated artifacts, as HiFi-SIM did with the best parameter ($\text{attStrength}=0.92$). We used NanoJ-SQUIRREL⁴ to calculate the signal error RSE between different reconstructions and widefield images, and the correlation coefficients (R^2 of linear fitting) between the incremental fluorescence intensity ΔF_{SIM} of SR reconstruction and the incremental $\Delta F_{\text{widefield}}$. The RSE of BF-SIM is smaller than that of HiFi-SIM, and the correlation coefficient (R^2) is also larger than that of HiFi-SIM. Therefore, even if the model may not apply to the background-free experiments, BF-SIM can still retain the signal, achieving a similar RSE and R squared compared with other methods.

Therefore, although we designed the BF background suppression model for background removal in the two-beam 2D-SIM and widefield microscopy, it may be prudently applied to image datasets of other microscopic modalities.

Response Fig. 2 | We applied the BF method in handling the 3D-SIM dataset. A fixed BSC-1C cell was immunostained with β -tubulin E7 and imaged under 3D-SIM (at a z-axial interval of 100 nm). **a**, The x-y (maximum projection), x-z, and y-z views of the SR volume were reconstructed plane-by-plane by three methods. **b**, The enlarged region of the yellow box in (a).

And we have incorporated parts of the discussion above in the **Supplementary Note 1 (Page 5, Lines 2-10)**.

Reviewer #3 (Remarks to the Author):

Mo and co-authors have devised a simple method to suppress out-of-focus background from raw 2D-SIM acquisitions, thereby achieving sectioning in a computational way. They demonstrate the usefulness of their computational sectioning method on several showcases, in particular on live actin, and even claim discovery in actin dynamics.

I think the basic idea is interesting and useful to the community, and will surely advance the potential of 2D SIM. I do, however, have several objections and questions that the authors should address.

Reply 1: *Thank you for commending our idea as "interesting and useful to the community, and will surely advance the potential of 2D SIM". We also thank you and other reviewers for the constructive suggestions, which motivate us to do more experiments, analyses, and explanations. This helps us to strengthen the manuscript substantially. With these modifications, we hope to convince you that we have adequately addressed all your concerns and suggestions.*

1. Incomplete information. The assessment of what is background in the raw images is done with a Fourier space filtering technique. Unfortunately, the information on the procedure is very sketchy. It is necessary to show the actual in and out-of-focus transfer function $H_{in}(k)$ and $H_{out}(k)$ and also the filter function $H_{out}/(H_{in} + H_{out})$ appearing in eq. 3. It is also necessary to show examples of several raw images and how they split into the background and foreground components according to the proposal. Also, the impact of the criterion what is termed in or out of focus should be investigated in detail. Now this is an arbitrary user defined 400 nm limit, which is a major weakness of the method. I would like to see the impact of variations of this number on the above mentioned filters and decompositions, as well as on the final reconstruction results. How else are we to know that we do not fall into the trap of "parameter tuning prone to human bias", which "fail at densely labeled fluorescent structures" that is the drawback of existing methods?

Reply 2: *Thank you and Review 1 for this important suggestion.*

*Regarding the definition of in-focus and out-of-focus range, as described in the **Supplementary material**, we consider signals within the z-axial resolution range in-focal signals, while signals outside the resolution range as out-of-focus background¹³. According to the Rayleigh criterion¹⁴, the z-axis resolution of the microscope is calculated as*

$$\Delta_{\text{Rayleigh},z} = \frac{2n\lambda}{NA^2}.$$

And the wavelength of the illumination laser used in our microscope system is 488~640 nm, and the effective NA of the objective lens corresponding to 2D-SIM is 1.4~1.5, so the average z-axis resolution is about 800 nm. That's why we set $\pm 0.4 \mu\text{m}$ as the in-focus signals ($h(\mathbf{r}, z_{in})$) originally. At the same time, the total intensity of PSF corresponding to the depth of $z = 4 \mu\text{m}$ is approximately 1% of that at $z = 0 \mu\text{m}$; thus, background at a depth of more than $4 \mu\text{m}$ is negligible. Thus we regard PSF in the range of $-4 \sim -0.4 \mu\text{m}$ and $0.4 \sim 4 \mu\text{m}$ to be out-of-focus signals $h(\mathbf{r}, z_{out})$.

To calculate the actual in-focus and out-of-focus optical transfer function $H_{in}(k)$ and $H_{out}(k)$, we simulate the 3D PSF of the microscope system, which is independent of the imaging sample and given by

$$h(x, y, z) = \left| C \int_0^1 J_0 \left[k \frac{NA}{n_i} (x^2 + y^2)^{0.5} \rho \right] e^{-\frac{1}{2} j k \rho^2 z \left(\frac{NA}{n_i} \right)^2} \rho d\rho \right|^2,$$

where C is a normalizing constant, J_0 is the Bessel function of the first kind of order zero, k is the wavenumber in

the vacuum of the emitted light, NA is the numerical aperture of the objective lens, n_i is the refractive index of the immersion layer and j is the imaginary number¹.

Based on this equation, we used a plugin PSF Generator of ImageJ to generate our 3D PSF². Therefore, we can easily pre-calculate $H_{in}(k)$ and $H_{out}(k)$. In other words, in the same microscope system, different samples are convolved with the same PSF to obtain the corresponding raw image. Therefore, we can calculate $H_{in}(k)$, $H_{out}(k)$, and the filtering function $H_{out}(k)/(H_{in}(k) + H_{out}(k))$. Per your suggestion, we have chosen different in-focal ranges ($\pm 0.2\mu\text{m}$, $\pm 0.4\mu\text{m}$, $\pm 0.6\mu\text{m}$, $\pm 0.8\mu\text{m}$). Thus we separated distributions of out-of-focus background and in-focus signal calculated from different samples with different $H_{in}(k)$ and $H_{out}(k)$ (**Supplementary Fig. 3**).

From this theory and its results, we conclude that $z_{in} = \pm 0.4\mu\text{m}$ is the ideal choice with a relatively small RSE, better background suppression, and good contrast. Therefore, except for the nonlinear GI-SIM and TIRF-SIM dataset (**Response Fig. 1**), we set the parameter to $\pm 0.4\mu\text{m}$ throughout the paper, including new experiments done during this round of revision.

2.Lack of comparison to state-of-the-art. The authors do show how their method can benefit a variety of SIM reconstruction methods, which is highly appreciated. The technique of notch filtering is known to suppress out-of-focus background from 2D-SIM reconstructions [Wicker et al. , Optics Express 21, 2032–2049, 2013; . Muller et al., Nature Communications 7, 10980, 2016]. This is clearly the state-of-the-art method that is closest in spirit to the proposed idea. Yet, a direct comparison to this method is missing, even though this is the most relevant comparison to make. In what ways is the proposed method better? Can more reliable reconstructions be guaranteed with the proposed method? Can the authors show a side-by-side comparison of the best possible background suppression that can be achieved with either method? How sensitive would these cases be to parameter selection? In my opinion, such a detailed comparison is necessary.

Reply 3: Thank you for your reminder. Although we did not directly compare BF-SIM with notch filtering previously, we did compare BF-SIM with the recently proposed HiFi-SIM, where HiFi-SIM incorporates notch filtering as the essential first step¹⁵. HiFi-SIM is a two-step SIM reconstruction algorithm with background suppression and spectrum optimization. Its first step is equivalent to the notch filtering to remove the background, while the second step compensates for removing weak signals by direct filtering in the first step¹⁵. According to all the comparisons conducted, our BF-SIM is superior to HiFi-SIM in terms of the integrity of fine structure maintenance, signal linearity, and parameter setting.

Per your suggestion, we have now compared BF-SIM side-by-side with the notch filtering method (NF-SIM, Muller et al., Nature Communications 7, 10980, 2016). The filter form adopted by NF-SIM is:

$$g(k) = 1 - \text{attstrength} \cdot \exp \left[-\frac{|k|^2}{2 \cdot \text{attwidth}^2} \right].$$

According to Müller et al., the range of attstrength shall be 0.95-0.999, which is critical for background suppression, and the range of attwidth shall be 0.8-2³. We set the attwidth to 2, and the attstrength to 0.99, 0.995, 0.997, or 0.999 for comparisons with BF-SIM.

From the actin filaments images we compared, $\text{attstrength}=0.99$ was ineffective in suppressing the background. At the same time, $\text{attstrength}=0.999$ demonstrated the best background suppression ability at the cost of losing intricate actin filaments (**Supplementary Fig. 6**). In that sense, $\text{attstrength}=0.995$ and 0.997 may be the better trade-offs. Nevertheless, actin filaments reconstructed by BF-SIM were always better contrasted than those processed with different attstrength parameters of the notch filtering (**Supplementary Fig. 6g**). In addition, we found that NF-SIM is very sensitive to parameter selection, as a fluctuation of attstrength of 0.01 (from 0.99 to 0.999) caused significant differences by the notch filtering. Besides, different image data require different optimal parameters of NF-SIM. For

example, $attstrength=0.999$ was sufficient in suppressing the background but removing weak signals for the actin filaments image (**Supplementary Fig. 6**). However, it was not enough to eliminate residual artifacts in reconstructing the image of the fluorescent beads (**Supplementary Fig. 5**). Therefore, our BF-SIM excels in suppressing background and retaining weak signals and superior in algorithm robustness and accessibility.

We have incorporated parts of the discussion above in the **manuscript (Page 4, Lines 25-30, Page 5, Lines 1-4)**.

3. Unsubstantiated claims.

a. The authors contrast their method to "selective frequency spectrum attenuation", but their background assessment is based on Fourier space filtering. Clearly, their method also qualifies as a "selective frequency spectrum attenuation" method, and should be named as such.

Reply 4: Thank you. We described HiFi-SIM or notch filtering here as a "selective frequency spectrum attenuation," meaning that these methods need to select parameters for attenuation based on the intensity and location of the "high-frequency spots" in the image. For example, according to Müller et al.³, the $attStrength$ is in the range of 0.95-0.999, and the $attwidth$ is within 0.8-2 for the following notch filter:

$$g(k) = 1 - attstrength \cdot \exp \left[-\frac{|k|^2}{2 \cdot attwidth^2} \right].$$

We used the term to emphasize that these methods require a parameter selection process rather than emphasizing that they compute in the frequency domain or the space domain. Because the model parameters of our BF-SIM are calculated according to the microscope system itself, it does not change as image data changes. To better describe the difference, we revise the "selective frequency spectrum attenuation" to "**parameter-selective frequency spectrum attenuation.**"

We have incorporated parts of the discussion above in the **manuscript (Page 2, Line 27, Page 7, Line 27)**.

b. The sub-70 nm resolution claim is a bit too thin to my mind. It is based on a best result measured in a patch in one image. The authors should replace this claim with typical or average FRC values they find in multiple patches in multiple images. Also, it is known that nonlinear processing techniques, such as deconvolution, can result in a better FRC resolution (higher precision), but can come at the expense of a loss in accuracy, as structures are presented in the reconstruction that do not represent reality (see e.g. ref 14). This makes the low sub-70 nm value perhaps not so valuable. A propos, one or more references to original work on FRC in electron and/or optical microscopy would also be appropriate.

Reply 5: We agree that we need to testify more rigorously about any resolution claim, which was why we measured FWHMs of actin filaments and the minimum FRC value to represent the resolution of the SR images. Previously we used a self-written PanelJ plugin for FRC calculations (Biorxiv⁹ and is under review in Nature Communications). Per your and **Reviewer 2's** suggestion, now we used NanoJ-SQUIRREL to re-analyze the images' minimum and mean FRC values (**Supplementary Fig. 11e and 13a**), which confirmed the results obtained with PanelJ (**Supplementary Fig. 11d**).

We have incorporated parts of the discussion above in the **manuscript (Page 6, Lines 13-15)**.

c. The key showcase on live actin (Fig. 4) is analyzed using a sparse deconvolution technique proposed by the same group (ref. 11). Unfortunately, this confounds their conclusions. Can the claimed effects on actin dynamics be seen owing to the sparse deconvolution technique, to the proposed background suppression method, or is a combination

of the two essential?

Reply 6: Thank you for your questions. Three examples of actin dynamics could also be distinguished under the conventional Wiener-SIM. As shown in **Response Fig. 3**, there were no statistical differences in the maximum average fluorescence intensity and fluorescence decay times of actin blip, cloud, and vortex among the Wiener-SIM, BF-SIM, and BF-Sparse-SIM reconstruction results. However, after the BF-Sparse deconvolution, the originally barely distinguishable vortex event became more prominent (**Response Fig. 3**).

On the other hand, we argued that improved resolution afforded by the BF-Sparse-SIM did lead to more resolvability. Per **Reviewer 2's** suggestion, we have imaged RAW264.7 cells with the SD-SIM and observed actin blips, clouds, and vortices in live cells. These events had maximal average intensities and decay kinetics similar to those yielded by BF-Sparse-SIM (**Supplementary Fig. 14**). However, we observed fewer vortex events with SD-SIM, and their sizes were larger. This is best explained by the lower resolution of SD-SIM compared to the BF-Sparse-SIM, which might only resolve large vortex structures and misclassify small ones as actin cloud events.

Response Fig. 3 | A representative example of an actin vortex event that increased fluorescence intensities followed the spiral and outward-to-inward pattern. **a** and **b**, Actin vortex examples of BF-Sparse-SIM and Wiener-SIM reconstruction results, and their original intensity trace (black) and the differentiated intensity trace (red), respectively.

The actual images shown with the sparse SIM technique do not seem artefact free to me. Ways to tackle these questions is to give FRC numbers with/without the background suppression method, and to do the same analysis on the live actin dataset by combining the method with standard SIM reconstruction, so as to eliminate deconvolution artefacts. If their conclusions on actin blips etc. then still stand it would make their cases stronger.

Reply 7: Thank you for this insightful suggestion. Therefore, we have analyzed the resolutions of Wiener-SIM without background suppression, Wiener-SIM with background suppression (BF-SIM), and Sparse-SIM. As the average FRC resolutions of Wiener-SIM and BF-SIM were indistinguishable (137.5 ± 1.5 nm and 137.7 ± 1.4 nm, **Supplementary Fig. 13** and **Supplementary Table.2**), the BF procedure did not sharpen the image. Only after the sparse deconvolution did we observe an increase in resolution, which agreed with our previous paper¹⁶. In addition, we

have re-calculated decay time constants and maximum intensities under different reconstructions (we did not re-calculate the event area as it depended on image resolution and contrast). Decay time constants and the maximum intensities of three types of actin events under the Wiener-SIM showed similar trends to those obtained by the BF-Sparse-SIM (**Supplementary Fig. 13b** versus **Fig. 4h**). Although the maximum intensity obtained by the Wiener-SIM were the most prominent (possibly due to the inclusion of integrated background fluorescence), three methods yield almost identical decay time constants (**Supplementary Fig. 13d**). These data reinforce the validity of BF-Sparse-SIM in reporting quantitatively accurate kinetics.

4. I would also like to note that I disagree with the data availability statement "upon reasonable request". I prefer fully open data, or a statement without this questionable disclaimer of "reasonable".

Reply 8: Agreed. We have uploaded raw images and reconstructed SR images from **Fig. 1, 2e, f, and Supplementary Fig. 3b,c, 6, 9, 10** at <https://doi.org/10.6084/m9.figshare.21792788>. In addition, we have also provided individual example datasets from **Figs. 3, 4, and Supplementary Fig. 12** there.

Finally, I cannot assess the credibility of the novelty claims made on actin dynamics, as I am no expert in actin biophysics.

Reply 9: As elaborated in **Reply 6**, we have shown three types of actin dynamics with two independent SR methods, confirming their existence in RAW264.7 cells. In addition, we did a literature search in PubMed for actin dynamics in macrophages and other cell types and failed to find any similar actin dynamics reported previously. Therefore, we have revised the statement in the manuscript as "**To our best knowledge,**".

We have incorporated parts of the discussion above in the **manuscript (Page 8, Line 23)**.

References

1. Aguet, F., Van De Ville, D. & Unser, M. Model-Based 2.5-D Deconvolution for Extended Depth of Field in Brightfield Microscopy. *IEEE Trans. Image Process.* **17**, 1144–1153 (2008).
2. Schneider, C. A., Rasband, W. S. & Eliceiri, K. W. NIH Image to ImageJ: 25 years of image analysis. *Nat. Methods* **9**, 671–675 (2012).
3. Müller, M., Mönkemöller, V., Hennig, S., Hübner, W. & Huser, T. Open-source image reconstruction of super-resolution structured illumination microscopy data in ImageJ. *Nat. Commun.* **7**, 10980 (2016).
4. Culley, S. *et al.* NanoJ-SQUIRREL: quantitative mapping and minimisation of super-resolution optical imaging artefacts. *Nat. Methods* **15**, 263–266 (2018).
5. Hebert, B., Costantino, S. & Wiseman, P. W. Spatiotemporal Image Correlation Spectroscopy (STICS) Theory, Verification, and Application to Protein Velocity Mapping in Living CHO Cells. *Biophys. J.* **88**, 3601–3614 (2005).
6. Ashdown, G. W. & Owen, D. M. Spatio-temporal image correlation spectroscopy and super-resolution microscopy to quantify molecular dynamics in T cells. *Methods* **140–141**, 112–118 (2018).
7. Ashdown, G., Pandžić, E., Cope, A., Wiseman, P. & Owen, D. Cortical Actin Flow in T Cells Quantified by Spatio-temporal Image Correlation Spectroscopy of Structured Illumination Microscopy Data. *J. Vis. Exp.* 53749 (2015) doi:10.3791/53749.
8. Zhang, Y. *et al.* Mitochondria determine the sequential propagation of the calcium macrodomains revealed by the super-resolution calcium lantern imaging. *Sci. China Life Sci.* **63**, 1543–1551 (2020).
9. Zhao, W. *et al.* Quantitatively mapping local quality of super-resolution microscopy by rolling Fourier ring correlation. 2022.12.01.518675 Preprint at <https://doi.org/10.1101/2022.12.01.518675> (2022).
10. Qiao, C. *et al.* Evaluation and development of deep neural networks for image super-resolution in optical

- microscopy. *Nat. Methods* **18**, 194–202 (2021).
11. Guo, Y. *et al.* Visualizing Intracellular Organelle and Cytoskeletal Interactions at Nanoscale Resolution on Millisecond Timescales. *Cell* **175**, 1430-1442.e17 (2018).
 12. Wen, G., Wang, L., Chen, X., Tang, Y. & Li, S. Frequency–spatial domain joint optimization for improving super-resolution images of nonlinear structured illumination microscopy. *Opt. Lett.* **46**, 5842–5845 (2021).
 13. Sandison, D. R. & Webb, W. W. Background rejection and signal-to-noise optimization in confocal and alternative fluorescence microscopes. *Appl. Opt.* **33**, 603–615 (1994).
 14. Rayleigh, Lord. V. *Investigations in optics, with special reference to the spectroscope.* Lond. Edinb. Dublin Philos. Mag. J. Sci. **9**, 40–55 (1880).
 15. Wen, G. *et al.* High-fidelity structured illumination microscopy by point-spread-function engineering. *Light Sci. Appl.* **10**, 70 (2021).
 16. Zhao, W. *et al.* Sparse deconvolution improves the resolution of live-cell super-resolution fluorescence microscopy. *Nat. Biotechnol.* **40**, 606–617 (2022).

Reviewers' comments:

Reviewer #1 (Remarks to the Author)

Additional comments :

Thanks for authors taking the comments into consideration and tremendous work to evident their BF-SIM principle and performance. With the correction in the main text and new materials in supplementary, in my opinion, the advantages of BF-SIM have been clearly demonstrated.

However, due to some misunderstandings of my questions, the paper can be further polished before publication.

1. I don't accept that approximation of equation P2 in general.

$$[g(\mathbf{r}, z_{out}) \cdot I(\mathbf{r})] \otimes h(\mathbf{r}, z_{out}) \approx [g(\mathbf{r}, z_{in}) \cdot I(\mathbf{r})] \otimes h(\mathbf{r}, z_{out}). \quad (\text{P2})$$

The authors added supplementary Fig.2 to demonstrate that the approximation is satisfied. But it was taken for a 3D cell sample. The fact that the out-of-focus images are the same for this sample doesn't mean that the approximation is true in general. For instance, taking again the single layer fluorescent beads sample, the $g(\mathbf{r}, z_{out})$ will be zero, hence the left part in equation P2 will be zero.

So I suggest the authors define the application scenarios for their model. And be more careful about using the word "physical model".

2. In supplementary Fig.2, did the images auto-scaled? Auto-scale should not be done and intensity scale bars should be given here.
3. I don't understand the supplementary Fig.3(a). Is the H_{in} (about 0.14) smaller than H_{out} (hard to see, about 0.25) even for a in-focus depth of $\pm 0.8\mu\text{m}$? And

I would expect that $H_{out}/(H_{in}+H_{out})$ will be decreased greatly with the increase of in-focus depth.

1. Incomplete information. The assessment of what is background in the raw images is done with a Fourier space filtering technique. Unfortunately, the information on the procedure is very sketchy. It is necessary to show the actual in and out-of-focus transfer function $H_{in}(k)$ and $H_{out}(k)$ and also the filter function $H_{out}/(H_{in}+H_{out})$ appearing in eq. 3. It is also necessary to show examples of several raw images and how they split into the background and foreground components according to the proposal. Also, the impact of the criterion what is termed in or out of focus should be investigated in detail. Now this is an arbitrary user defined .. nm limit, which is a major weakness of the method. I would like to see the impact of variations of this number on the above mentioned filters and decompositions, as well as on the final reconstruction results. How else are we to know that we do not fall into the trap of "parameter tuning prone to human bias", which "fail at densely labeled fluorescent structures" that is the drawback of existing methods?

Reply 2: Thank you and **Review 1** for this important suggestion.

Regarding the definition of in-focus and out-of-focus range, as described in the **Supplementary material**, we consider signals within the z-axial resolution range in-focal signals, while signals outside the resolution range as out-of-focus background¹³. According to the Rayleigh criterion¹⁴, the z-axis resolution of the microscope is calculated as $\Delta_{Rayleigh,z} = 2n\lambda NA^2$.

And the wavelength of the illumination laser used in our microscope system is 488~640 nm, and the effective NA of the objective lens corresponding to 2D-SIM is 1.4~1.5, so the average z-axis resolution is about 800 nm. That's why we set $\pm 0.4 \mu m$ as the in-focus signals ($h(\mathbf{r}, z_{in})$) originally. At the same time, the total intensity of PSF corresponding to the depth of $z = 4 \mu m$ is approximately 1% of that at $z = 0 \mu m$; thus, background at a depth of more than $4 \mu m$ is negligible. Thus we regard PSF in the range of $-4 \sim -0.4 \mu m$ and $0.4 \sim 4 \mu m$ to be out-of-focus signals $h(\mathbf{r}, z_{out})$.

To calculate the actual in-focus and out-of-focus optical transfer function $H_{in}(k)$ and $H_{out}(k)$, we simulate the 3D PSF of the microscope system, which is independent of the imaging sample and given by $h(x, y, z) = |C \int_0^{\infty} [kNA n_i (x^2 + y^2)^{0.5} \rho] e^{-12jk\rho z (NA n_i)^2 \rho d\rho} |^2$,

where C is a normalizing constant, J_0 is the Bessel function of the first kind of order zero, k is the wavenumber in the vacuum of the emitted light, NA is the numerical aperture of the objective lens, n_i is the refractive index of the immersion layer and j is the imaginary number¹.

Based on this equation, we used a plugin PSF Generator of ImageJ to generate our 3D PSF².

Therefore, we can easily pre-calculate $H_{in}(k)$ and $H_{out}(k)$. In other words, in the same microscope system, different samples are convolved with the same PSF to obtain the corresponding raw image. Therefore, we can calculate $H_{in}(k)$, $H_{out}(k)$, and the filtering function $H_{out}(k)/(H_{in}(k)+H_{out}(k))$. Per your suggestion, we have chosen different in-focal ranges ($\pm 0.2 \mu m$, $\pm 0.4 \mu m$, $\pm 0.6 \mu m$, $\pm 0.8 \mu m$). Thus we separated distributions of out-of-focus background and in-focus signal calculated from different samples with different $H_{in}(k)$ and $H_{out}(k)$

(**Supplementary Fig. 3**).

From this theory and its results, we conclude that $z_{in} = \pm 0.4 \mu m$ is the ideal choice with a relatively small RSE, better background suppression, and good contrast. Therefore, except for the nonlinear GI-SIM and TIRF-SIM dataset (**Response Fig. 1**), we set the parameter to $\pm 0.4 \mu m$ throughout the paper, including new experiments done during this round of revision.

My comment: the authors calculated the $H_{in}(k)$ and $H_{out}(k)$ based on the 3D PSF model. Based on the calculation, the z range for defocus is rationalized. I am satisfied with the authors' reply to

this comment.

2. Lack of comparison to state-of-the-art. The authors do show how their method can benefit a variety of SIM reconstruction methods, which is highly appreciated. The technique of notch filtering is known to suppress out-of-focus background from 2D-SIM reconstructions [Wicker et al., Optics Express 21, 2.32–2.9, 2.13; Müller et al., Nature Communications 7, 1.98., 2.16]. This is clearly the state-of-the-art method that is closest in spirit to the proposed idea. Yet, a direct comparison to this method is missing, even though this is the most relevant comparison to make. In what ways is the proposed method better? Can more reliable reconstructions be guaranteed with the proposed method? Can the authors show a side-by-side comparison of the best possible background suppression that can be achieved with either method? How sensitive would these cases be to parameter selection? In my opinion, such a detailed comparison is necessary.

Reply 3: *Thank you for your reminder. Although we did not directly compare BF-SIM with notch filtering previously, we did compare BF-SIM with the recently proposed HiFi-SIM, where HiFi-SIM incorporates notch filtering as the essential first step¹⁵. HiFi-SIM is a two-step SIM reconstruction algorithm with background suppression and spectrum optimization. Its first step is equivalent to the notch filtering to remove the background, while the second step compensates for removing weak signals by direct filtering in the first step¹⁵. According to all the comparisons conducted, our BF-SIM is superior to HiFi-SIM in terms of the integrity of fine structure maintenance, signal linearity, and parameter setting.*

Per your suggestion, we have now compared BF-SIM side-by-side with the notch filtering method (NF-SIM, Müller et al., Nature Communications 7, 10980, 2016). The filter form adopted by NF-SIM is: $g(k) = 1 - \text{attstrength} \cdot \exp[-|k|^2 \cdot \text{attwidth}^2]$.

According to Müller et al., the range of attstrength shall be 0.95–0.999, which is critical for background suppression, and the range of attwidth shall be 0.8–2.3. We set the attwidth to 2, and the attstrength to 0.99, 0.995, 0.997, or 0.999 for comparisons with BF-SIM.

*From the actin filaments images we compared, attstrength=0.99 was ineffective in suppressing the background. At the same time, attstrength=0.999 demonstrated the best background suppression ability at the cost of losing intricate actin filaments (**Supplementary Fig. 6**). In that sense, attstrength=0.995 and 0.997 may be the better trade-offs. Nevertheless, actin filaments reconstructed by BF-SIM were always better contrasted than those processed with different attstrength parameters of the notch filtering (**Supplementary Fig. 6g**). In addition, we found that NF-SIM is very sensitive to parameter selection, as a fluctuation of attstrength of 0.01 (from 0.99 to 0.999) caused significant differences by the notch filtering. Besides, different image data require different optimal parameters of NF-SIM. For 14*

example, $attstrength=0.999$ was sufficient in suppressing the background but removing weak signals for the actin filaments image (**Supplementary Fig. 6**). However, it was not enough to eliminate residual artifacts in reconstructing the image of the fluorescent beads (**Supplementary Fig. 5**). Therefore, our BF-SIM excels in suppressing background and retaining weak signals and superior in algorithm robustness and accessibility.

We have incorporated parts of the discussion above in the **manuscript (Page 4, Lines 25-30, Page 5, Lines 1-4)**.

My comments: The authors made detailed comparisons and added the results in the supplementary. I would say that the superiority of BF-SIM has been demonstrated in their claimed scenarios.

3. Unsubstantiated claims.

a. The authors contrast their method to "selective frequency spectrum attenuation", but their background assessment is based on Fourier space filtering. Clearly, their method also qualifies as a "selective frequency spectrum attenuation" method, and should be named as such.

Reply 4: Thank you. We described HiFi-SIM or notch filtering here as a "selective frequency spectrum attenuation," meaning that these methods need to select parameters for attenuation based on the intensity and location of the "high-frequency spots" in the image. For example, according to Müller et al.³, the $attStrength$ is in the range of 0.95-0.999, and the $attwidth$ is within 0.8-2 for the following notch filter: $g(k)=1-attstrength \cdot \exp[-|k|^2 \cdot attwidth^2]$.

We used the term to emphasize that these methods require a parameter selection process rather than emphasizing that they compute in the frequency domain or the space domain. Because the model parameters of our BF-SIM are calculated according to the microscope system itself, it does not change as image data changes. To better describe the difference, we revise the "selective frequency spectrum attenuation" to "**parameter-selective frequency spectrum attenuation**."

We have incorporated parts of the discussion above in the **manuscript (Page 2, Line 27, Page 7, Line 27)**.

My comments: I agree on the revision from "selective frequency spectrum attenuation" to "parameter-selective frequency spectrum attenuation". However, in BF-SIM, Wiener-SIM is used for reconstruction after background removal, and there is attenuation in Wiener-SIM. The authors may classify this point also.

b. The sub-7. nm resolution claim is a bit too thin to my mind. It is based on a best result measured in a patch in one image. The authors should replace this claim with typical or average FRC values they find in multiple patches in multiple images. Also, it is known that nonlinear processing techniques, such as deconvolution, can result in a better FRC resolution (higher precision), but can come at the expense of a loss in accuracy, as structures are presented in the reconstruction that do not represent reality (see e.g. ref 1). This makes the low sub-7. nm value perhaps not so valuable. A propos, one or more references to original work on FRC in electron and/or optical microscopy would also be appropriate.

Reply 5: We agree that we need to testify more rigorously about any resolution claim, which was why we measured FWHMs of actin filaments and the minimum FRC value to represent the resolution of the SR images. Previously we used a self-written PanelJ plugin for FRC calculations (Biorxiv⁹ and is under review in Nature Communications). Per your and **Reviewer 2's** suggestion,

now we used NanoJ-SQUIRREL to re-analyze the images' minimum and mean FRC values (**Supplementary Fig. 11e and 13a**), which confirmed the results obtained with Panel J (**Supplementary Fig. 11d**).

We have incorporated parts of the discussion above in the **manuscript (Page 6, Lines 13-15)**.

My comment: I am satisfied with the revised resolution calculation.

c. The key showcase on live actin (Fig.) is analyzed using a sparse deconvolution technique proposed by the same group (ref. 11). Unfortunately, this confounds their conclusions. Can the claimed effects on actin dynamics be seen owing to the sparse deconvolution technique, to the proposed background suppression method, or is a combination of the two essential?

Reply 6: Thank you for your questions. Three examples of actin dynamics could also be distinguished under the conventional Wiener-SIM. As shown in **Response Fig. 3**, there were no statistical differences in the maximum average fluorescence intensity and fluorescence decay times of actin blip, cloud, and vortex among the Wiener-SIM, BF-SIM, and BF-Sparse-SIM reconstruction results. However, after the BF-Sparse deconvolution, the originally barely distinguishable vortex event became more prominent (**Response Fig. 3**).

On the other hand, we argued that improved resolution afforded by the BF-Sparse-SIM did lead to more resolvability. Per **Reviewer 2's** suggestion, we have imaged RAW264.7 cells with the SD-SIM and observed actin blips, clouds, and vortexes in live cells. These events had maximal average intensities and decay kinetics similar to those yielded by BF-Sparse-SIM (**Supplementary Fig. 14**). However, we observed fewer vortex events with SD-SIM, and their sizes were larger. This is best explained by the lower resolution of SD-SIM compared to the BF-Sparse-SIM, which might only resolve large vortex structures and misclassify small ones as actin cloud events.

My comment: I agree that the vortex event became more prominent with BF-Sparse-SIM. But still, the difference is to some extent, not essential, I would say.

The actual images shown with the sparse SIM technique do not seem artefact free to me. Ways to tackle these questions is to give FRC numbers with/without the background suppression method, and to do the same analysis on the live actin dataset by combining the method with standard SIM reconstruction, so as to eliminate deconvolution artefacts. If their conclusions on actin blips etc. then still stand it would make their cases stronger.

Reply 7: Thank you for this insightful suggestion. Therefore, we have analyzed the resolutions of Wiener-SIM without background suppression, Wiener-SIM with background suppression (BF-SIM), and Sparse-SIM. As the average FRC resolutions of Wiener-SIM and BF-SIM were indistinguishable (137.5 ± 1.5 nm and 137.7 ± 1.4 nm, **Supplementary Fig. 13 and Supplementary Table.2**), the BF procedure did not sharpen the image. Only after the sparse deconvolution did we observe an increase in resolution, which agreed with our previous paper. In addition, we have re-calculated decay time constants and maximum intensities under different reconstructions (we did not re-calculate the event area as it depended on image resolution and contrast). Decay time constants and the maximum intensities of three types of actin events under the Wiener-SIM showed similar trends to those obtained by the BF- Sparse-SIM (**Supplementary Fig. 13b versus Fig. 4h**). Although the maximum intensity obtained by the Wiener-SIM were the

most prominent (possibly due to the inclusion of integrated background fluorescence), three methods yield almost identical decay time constants (Supplementary Fig. 13d). These data reinforce the validity of BF-Sparse-SIM in reporting quantitatively accurate kinetics.

My comment: The author calculated the FRC numbers as suggested. However, I didn't see improved image quality with the BF procedure from these values.

. I would also like to note that I disagree with the data availability statement "upon reasonable request". I prefer fully open data, or a statement without this questionable disclaimer of "reasonable".

*Reply 8: Agreed. We have uploaded raw images and reconstructed SR images from **Fig. 1, 2e, f,** and **Supplementary Fig. 3b,c, 6, 9, 10** at <https://doi.org/10.6084/m9.figshare.21792788>. In addition, we have also provided individual example datasets from **Figs. 3, 4,** and **Supplementary Fig. 12** there.*

My comment: Thanks for the open source data.

Finally, I cannot assess the credibility of the novelty claims made on actin dynamics, as I am no expert in actin biophysics.

*Reply 9: As elaborated in **Reply 6**, we have shown three types of actin dynamics with two independent SR methods, confirming their existence in RAW264.7 cells. In addition, we did a literature search in PubMed for actin dynamics in macrophages and other cell types and failed to find any similar actin dynamics reported previously. Therefore, we have revised the statement in the manuscript as "**To our best knowledge,**".*

*We have incorporated parts of the discussion above in the **manuscript (Page 8, Line 23)**.*

My comment: OK.

Reviewer #2 (Remarks to the Author)

The authors have addressed all my comments. I now recommend publication—congratulation on a nice paper.

RESPONSE TO REVIEWERS' COMMENTS

We would like to express our gratitude to all the reviewers for providing feedback on our manuscript and our previous point-to-point response (NCOMMS-22-31115A), especially Reviewer #2 for his valuable input. In response to the new questions raised by the reviewer regarding the principle of the BF model, we have included **2 new Response Figures (Fig. P1, P2)** in the Point-to-point responses and made modifications to **2 new Supplementary Figures (2, 3)** to address the concerns comprehensively. We have also provided further explanations and discussions on some expression problems in the manuscript and incorporated the reviewer's suggestions for revisions. Additionally, to demonstrate the superiority of our data processing process for the BF-Sparse SIM analysis of actin dynamics in living cells, we have included a new **new Response Figure (Fig. P3)**.

In this document, comments made by the reviewers are shown in black and red fonts, our previous responses are displayed in blue italic fonts, and our updated response is shown in green italic fonts. Furthermore, we have carefully revised the manuscript based on editorial suggestions (author checklist document), added more material and methods, and included necessary information missing from previous figures. These revisions are highlighted in red in the manuscript and supplementary information. Through these modifications, we hope to provide sufficient evidence to demonstrate the usefulness and superiority of BF-SIM and BF-Sparse-SIM, and we sincerely hope that you will agree to its publication in *Nature Communications*.

Additional comments :

Thanks for authors taking the comments into consideration and tremendous work to evident their BF-SIM principle and performance. With the correction in the main text and new materials in supplementary, in my opinion, the advantages of BF-SIM have been clearly demonstrated.

However, due to some misunderstandings of my questions, the paper can be further polished before publication.

1. I don't accept that approximation of equation P2 in general.

$$[g(\mathbf{r}, z_{out}) \cdot I(\mathbf{r})] \otimes h(\mathbf{r}, z_{out}) \approx [g(\mathbf{r}, z_{in}) \cdot I(\mathbf{r})] \otimes h(\mathbf{r}, z_{out}). \quad (\text{P2})$$

The authors added supplementary Fig.2 to demonstrate that the approximation is satisfied. But it was taken for a 3D cell sample. The fact that the out-of-focus images are the same for this sample doesn't mean that the approximation is true in general. For instance, taking again the single layer fluorescent beads sample, the $g(\mathbf{r}, z_{out})$ will be zero, hence the left part in equation P2 will be zero.

So I suggest the authors define the application scenarios for their model. And be more careful about using the word "physical model".

Reply 1: Thank you for your suggestion. As previously explained in our proposed BF model, BF-SIM is suitable for 2D-SIM superresolution imaging of living cells. We have now emphasized the application scenarios of BF in both the **abstract** and **conclusion** of the manuscript (**Page 2, Line 7; Page 7, Line 28**). In addition, we also wanted to add a description to the title, but due to the word limit of the title, we gave up this option.

As you said, when the image condition does not satisfy the premise of BF, the calculation result may subtract the fluorescence that should not be subtracted. We have also carried out BF preprocessing in nonlinear GI-SIM (previous response to Reviewer #2), 3D-SIM (previous response to Reviewer #3), beads, and other cases that do not conform to the hypothesis. The results showed that BF did not adversely affect the data. We have re-displayed the intensity scale in the previous images of beads (**Fig. P1**). This figure demonstrates that BF does not significantly alter the contrast of beads, and any fluorescence that was incorrectly removed appeared to be negligible. Of course, we cannot guarantee that our algorithm will never incorrectly remove information, as there is no ground truth for real-world image data. As such, we recommend using our method as a starting point and then fine-tuning the results as needed based on personal preferences.

Finally, we utilize the "physical model" to emphasize the fundamental contrast between our BF model, which is motivated by the physical model of the actual imaging process, and the traditional artificial attenuation of a region of the Wiener-SIM spectrum. As a result, we still hope to use a representation like a "physical model."

Fig. P1 | SR images of fluorescence beads with and without BF preprocessing with different SNR. Fluorescence beads with diameters of 200 nm were observed under the high SNR (upper) and low SNR (bottom). The images from left to right are Wiener-SIM without background suppression, BF-SIM with background suppression, and residuals of Wiener-SIM and BF-SIM. All images are shown on the same intensity scale.

2. In supplementary Fig.2, did the images auto-scaled? Auto-scale should not be done and intensity scale bars should be given here.

Reply 2: Thank you for your suggestion. The images in **Supplementary Fig.2** all use the same intensity scale, and we now add intensity scale bars in the figure.

3. I don't understand the supplementary Fig.3(a). Is the H_{in} (about 0.14) smaller than H_{out} (hard to see, about 0.25) even for a in-focus depth of $\pm 0.8 \mu\text{m}$? And I would expect that $H_{out}/(H_{in}+H_{out})$ will be decreased greatly with the increase of in-focus depth.

Reply 3: Thank you for your inquiries. We apologize for any inconvenience caused by the unclear display of the previous **Supplementary Fig.3a**. We have re-drawn the distribution of H_{in} , H_{out} , and $H_{out}/(H_{in}+H_{out})$, which are now presented in **Fig. P2**. Our findings show that the peak value of H_{out} is higher than that of H_{in} when the in-focus depth is $\pm 0.2 \mu\text{m}$, $\pm 0.4 \mu\text{m}$, $\pm 0.6 \mu\text{m}$, and $\pm 0.8 \mu\text{m}$, respectively. As the in-focus depth increases, the peak values of $H_{out}/(H_{in}+H_{out})$ indeed decrease, as you pointed out. However, their distributions remain similar without a significant decline. We have updated the relevant data in **Supplementary Fig.3a**.

Fig. P2 | The H_{in} , H_{out} and $H_{out}/(H_{in} + H_{out})$ distributions when the in-focus depth $z_{in} = \pm 0.2 \mu\text{m}$, $\pm 0.4 \mu\text{m}$, $\pm 0.6 \mu\text{m}$, $\pm 0.8 \mu\text{m}$, respectively.

Previous comments :

1. Incomplete information. The assessment of what is background in the raw images is done with a Fourier space filtering technique. Unfortunately, the information on the procedure is very sketchy. It is necessary to show the actual in and out-of-focus transfer function $H_{in}(k)$ and $H_{out}(k)$ and also the filter function $H_{out}/(H_{in} + H_{out})$ appearing in eq. 3. It is also necessary to show examples of several raw images and how they split into the background and foreground components according to the proposal. Also, the impact of the criterion what is termed in or out of focus should be investigated in detail. Now this is an arbitrary user defined 400 nm limit, which is a major weakness of the method. I would like to see the impact of variations of this number on the above mentioned filters and decompositions, as well as on the final reconstruction results. How else are we to know that we do not fall into the trap of "parameter tuning prone to human bias", which "fail at densely labeled fluorescent structures" that is the drawback of existing methods?

Reply 2: Thank you and **Review 1** for this important suggestion.

Regarding the definition of in-focus and out-of-focus range, as described in the **Supplementary material**, we consider signals within the z-axial resolution range in-focal signals, while signals outside the resolution range as out-of-focus background¹³. According to the Rayleigh criterion¹⁴, the z-axis resolution of the microscope is calculated as

$$\Delta_{\text{Rayleigh},z} = \frac{2n\lambda}{NA^2}.$$

And the wavelength of the illumination laser used in our microscope system is 488–640 nm, and the effective NA of the objective lens corresponding to 2D-SIM is 1.4–1.5, so the average z-axis resolution is about 800 nm. That's why we set $\pm 0.4 \mu\text{m}$ as the in-focus signals ($h(\mathbf{r}, z_{in})$) originally. At the same time, the total intensity of PSF corresponding to the depth of $z = 4 \mu\text{m}$ is approximately 1% of that at $z = 0 \mu\text{m}$; thus, background at a depth of more than $4 \mu\text{m}$ is negligible. Thus we regard PSF in the range of $-4 \sim -0.4 \mu\text{m}$ and $0.4 \sim 4 \mu\text{m}$ to be out-of-focus signals $h(\mathbf{r}, z_{out})$.

To calculate the actual in-focus and out-of-focus optical transfer function $H_{in}(k)$ and $H_{out}(k)$, we simulate the 3D PSF of the microscope system, which is independent of the imaging sample and given by

$$h(x, y, z) = \left| C \int_0^1 J_0 \left[k \frac{NA}{n_i} (x^2 + y^2)^{0.5} \rho \right] e^{-\frac{1}{2} j k \rho^2 z \left(\frac{NA}{n_i} \right)^2} \rho d\rho \right|^2,$$

where C is a normalizing constant, J_0 is the Bessel function of the first kind of order zero, k is the wavenumber in the vacuum of the emitted light, NA is the numerical aperture of the objective lens, n_i is the refractive index of the immersion layer and j is the imaginary number¹.

Based on this equation, we used a plugin PSF Generator of ImageJ to generate our 3D PSF². Therefore, we can easily pre-calculate $H_{in}(k)$ and $H_{out}(k)$. In other words, in the same microscope system, different samples are convolved with the same PSF to obtain the corresponding raw image. Therefore, we can calculate $H_{in}(k)$, $H_{out}(k)$, and the filtering function $H_{out}(k)/(H_{in}(k) + H_{out}(k))$. Per your suggestion, we have chosen different in-focal ranges ($\pm 0.2 \mu\text{m}$, $\pm 0.4 \mu\text{m}$, $\pm 0.6 \mu\text{m}$, $\pm 0.8 \mu\text{m}$). Thus we separated distributions of out-of-focus background and in-focus signal calculated from different samples with different $H_{in}(k)$ and $H_{out}(k)$ (**Supplementary Fig. 3**).

From this theory and its results, we conclude that $z_{in} = \pm 0.4 \mu\text{m}$ is the ideal choice with a relatively small RSE, better background suppression, and good contrast. Therefore, except for the nonlinear GI-SIM and TIRF-SIM dataset (**Response Fig. 1**), we set the parameter to $\pm 0.4 \mu\text{m}$ throughout the paper, including new experiments done during this round of revision.

My comment: the authors calculated the $H_{in}(k)$ and $H_{out}(k)$ based on the 3D PSF model. Based on the calculation, the

z range for defocus is rationalized. I am satisfied with the authors' reply to this comment.

Reply: Thank you.

2. Lack of comparison to state-of-the-art. The authors do show how their method can benefit a variety of SIM reconstruction methods, which is highly appreciated. The technique of notch filtering is known to suppress out-of-focus background from 2D-SIM reconstructions [Wicker et al., *Optics Express* 21, 2032–2049, 2013; Müller et al., *Nature Communications* 7, 10980, 2016]. This is clearly the state-of-the-art method that is closest in spirit to the proposed idea. Yet, a direct comparison to this method is missing, even though this is the most relevant comparison to make. In what ways is the proposed method better? Can more reliable reconstructions be guaranteed with the proposed method? Can the authors show a side-by-side comparison of the best possible background suppression that can be achieved with either method? How sensitive would these cases be to parameter selection? In my opinion, such a detailed comparison is necessary.

Reply 3: Thank you for your reminder. Although we did not directly compare BF-SIM with notch filtering previously, we did compare BF-SIM with the recently proposed HiFi-SIM, where HiFi-SIM incorporates notch filtering as the essential first step¹⁵. HiFi-SIM is a two-step SIM reconstruction algorithm with background suppression and spectrum optimization. Its first step is equivalent to the notch filtering to remove the background, while the second step compensates for removing weak signals by direct filtering in the first step¹⁵. According to all the comparisons conducted, our BF-SIM is superior to HiFi-SIM in terms of the integrity of fine structure maintenance, signal linearity, and parameter setting.

Per your suggestion, we have now compared BF-SIM side-by-side with the notch filtering method (NF-SIM, Müller et al., *Nature Communications* 7, 10980, 2016). The filter form adopted by NF-SIM is:

$$g(k) = 1 - \text{attstrength} \cdot \exp\left[-\frac{|k|^2}{2 \cdot \text{attwidth}^2}\right].$$

According to Müller et al., the range of attstrength shall be 0.95–0.999, which is critical for background suppression, and the range of attwidth shall be 0.8–2³. We set the attwidth to 2, and the attstrength to 0.99, 0.995, 0.997, or 0.999 for comparisons with BF-SIM.

From the actin filaments images we compared, attstrength=0.99 was ineffective in suppressing the background. At the same time, attstrength=0.999 demonstrated the best background suppression ability at the cost of losing intricate actin filaments (**Supplementary Fig. 6**). In that sense, attstrength=0.995 and 0.997 may be the better trade-offs. Nevertheless, actin filaments reconstructed by BF-SIM were always better contrasted than those processed with different attstrength parameters of the notch filtering (**Supplementary Fig. 6g**). In addition, we found that NF-SIM is very sensitive to parameter selection, as a fluctuation of attstrength of 0.01 (from 0.99 to 0.999) caused significant differences by the notch filtering. Besides, different image data require different optimal parameters of NF-SIM. For example, attstrength=0.999 was sufficient in suppressing the background but removing weak signals for the actin filaments image (**Supplementary Fig. 6**). However, it was not enough to eliminate residual artifacts in reconstructing the image of the fluorescent beads (**Supplementary Fig. 5**). Therefore, our BF-SIM excels in suppressing background and retaining weak signals and superior in algorithm robustness and accessibility.

We have incorporated parts of the discussion above in the **manuscript (Page 4, Lines 25–30, Page 5, Lines 1–4)**.

My comments: The authors made detailed comparisons and added the results in the supplementary. I would say that the superiority of BF-SIM has been demonstrated in their claimed scenarios.

Reply: Thank you.

3. Unsubstantiated claims.

a. The authors contrast their method to "selective frequency spectrum attenuation", but their background assessment is based on Fourier space filtering. Clearly, their method also qualifies as a "selective frequency spectrum attenuation" method, and should be named as such.

Reply 4: Thank you. We described HiFi-SIM or notch filtering here as a "selective frequency spectrum attenuation," meaning that these methods need to select parameters for attenuation based on the intensity and location of the "high-frequency spots" in the image. For example, according to Müller et al.³, the attStrength is in the range of 0.95-0.999, and the attwidth is within 0.8-2 for the following notch filter:

$$g(k) = 1 - \text{attstrength} \cdot \exp \left[-\frac{|k|^2}{2 \cdot \text{attwidth}^2} \right].$$

We used the term to emphasize that these methods require a parameter selection process rather than emphasizing that they compute in the frequency domain or the space domain. Because the model parameters of our BF-SIM are calculated according to the microscope system itself, it does not change as image data changes. To better describe the difference, we revise the "selective frequency spectrum attenuation" to "parameter-selective frequency spectrum attenuation."

We have incorporated parts of the discussion above in the **manuscript (Page 2, Line 27, Page 7, Line 27)**.

My comments: I agree on the revision from "selective frequency spectrum attenuation" to "parameter-selective frequency spectrum attenuation". However, in BF-SIM, Wiener-SIM is used for reconstruction after background removal, and there is attenuation in Wiener-SIM. The authors may classify this point also.

Reply: Thank you. From the perspective of suppressing low-frequency background components, the effect of BF is equivalent to attenuation by a Notch filter in Wiener-SIM. However, the BF method and the Notch filter differ in their approach to solving background problems. The BF method is inspired by the physical model of the imaging process to remove the background, while previous methods like the Notch filter directly attenuate a particular region of the Wiener-SIM spectrum. We have incorporated this point in the **manuscript (Page 7, Lines 28-32)**.

b. The sub-70 nm resolution claim is a bit too thin to my mind. It is based on a best result measured in a patch in one image. The authors should replace this claim with typical or average FRC values they find in multiple patches in multiple images. Also, it is known that nonlinear processing techniques, such as deconvolution, can result in a better FRC resolution (higher precision), but can come at the expense of a loss in accuracy, as structures are presented in the reconstruction that do not represent reality (see e.g. ref 14). This makes the low sub-70 nm value perhaps not so valuable. A propos, one or more references to original work on FRC in electron and/or optical microscopy would also be appropriate.

Reply 5: We agree that we need to testify more rigorously about any resolution claim, which was why we measured FWHMs of actin filaments and the minimum FRC value to represent the resolution of the SR images. Previously we used a self-written PanelJ plugin for FRC calculations (Biorxiv⁹ and is under review in Nature Communications). Per your and **Reviewer 2's** suggestion, now we used NanoJ-SQUIRREL to re-analyze the images' minimum and mean FRC values (**Supplementary Fig. 11e and 13a**), which confirmed the results obtained with PanelJ (**Supplementary Fig. 11d**).

We have incorporated parts of the discussion above in the **manuscript (Page 6, Lines 13-15)**.

My comment: I am satisfied with the revised resolution calculation.

Reply: Thank you.

c. The key showcase on live actin (Fig. 4) is analyzed using a sparse deconvolution technique proposed by the same group (ref. 11). Unfortunately, this confounds their conclusions. Can the claimed effects on actin dynamics be seen owing to the sparse deconvolution technique, to the proposed background suppression method, or is a combination of the two essential?

Reply 6: Thank you for your questions. Three examples of actin dynamics could also be distinguished under the conventional Wiener-SIM. As shown in Response Fig. 3, there were no statistical differences in the maximum average fluorescence intensity and fluorescence decay times of actin blip, cloud, and vortex among the Wiener-SIM, BF-SIM, and BF-Sparse-SIM reconstruction results. However, after the BF-Sparse deconvolution, the originally barely distinguishable vortex event became more prominent (Response Fig. 3).

On the other hand, we argued that improved resolution afforded by the BF-Sparse-SIM did lead to more resolvability. Per Reviewer 2's suggestion, we have imaged RAW264.7 cells with the SD-SIM and observed actin blips, clouds, and vortexes in live cells. These events had maximal average intensities and decay kinetics similar to those yielded by BF-Sparse-SIM (Supplementary Fig. 14). However, we observed fewer vortex events with SD-SIM, and their sizes were larger. This is best explained by the lower resolution of SD-SIM compared to the BF-Sparse-SIM, which might only resolve large vortex structures and misclassify small ones as actin cloud events.

Response Fig. 3 | A representative example of an actin vortex event that increased fluorescence intensities followed the spiral and outward-to-inward pattern. a and b, Actin vortex examples of BF-Sparse-SIM and Wiener-SIM reconstruction results, and their original intensity trace (black) and the differentiated intensity trace (red), respectively.

My comment: I agree that the vortex event became more prominent with BF-Sparse-SIM. But still, the difference is to some extent, not essential, I would say.

Reply: Thank you for your comment. Since BF-Sparse-SIM is a high-fidelity processing method, we anticipate that it will yield strong fluorescence signal distributions comparable to those achieved with Wiener-SIM. However, distinctions in weak signals may be observed. As illustrated in Fig. P3, a weakly fluorescent event is evident in which

the actin vortex fluorescence intensity increased from the outside to the inside in BF-Sparse-SIM, as depicted by the yellow dashed areas marked in the 0s, 2s, and 4s images. In contrast, this process is difficult to discern in the Wiener-SIM image. It is worth noting that this event could be mistaken for fluorescence emanating from the position indicated by the red arrow and diffusing to the surrounding area, resulting in its misclassification as the "actin cloud" category. Therefore, we believe that BF-Sparse-SIM shall be beneficial in accurately observing the intricate dynamics of living cells.

Fig. P3 | *A representative example of the fluorescence ascending process of a weak signal actin vortex. The upper images were obtained using BF-Sparse-SIM, while the lower images were acquired using Wiener-SIM. The yellow dashed lines indicate the area where the fluorescence intensity increases from the outside to the inside, and the red arrow points to the region with a strong signal in the images.*

The actual images shown with the sparse SIM technique do not seem artefact free to me. Ways to tackle these questions is to give FRC numbers with/without the background suppression method, and to do the same analysis on the live actin dataset by combining the method with standard SIM reconstruction, so as to eliminate deconvolution artefacts. If their conclusions on actin blips etc. then still stand it would make their cases stronger.

Reply 7: *Thank you for this insightful suggestion. Therefore, we have analyzed the resolutions of Wiener-SIM without background suppression, Wiener-SIM with background suppression (BF-SIM), and Sparse-SIM. As the average FRC resolutions of Wiener-SIM and BF-SIM were indistinguishable (137.5 ± 1.5 nm and 137.7 ± 1.4 nm, **Supplementary Fig. 13** and **Supplementary Table.2**), the BF procedure did not sharpen the image. Only after the sparse deconvolution did we observe an increase in resolution, which agreed with our previous paper¹⁶. In addition, we have re-calculated decay time constants and maximum intensities under different reconstructions (we did not re-calculate the event area as it depended on image resolution and contrast). Decay time constants and the maximum intensities of three types of actin events under the Wiener-SIM showed similar trends to those obtained by the BF-Sparse-SIM (**Supplementary Fig. 13b** versus **Fig. 4h**). Although the maximum intensity obtained by the Wiener-SIM were the most prominent (possibly due to the inclusion of integrated background fluorescence), three methods yield almost identical decay time constants (**Supplementary Fig. 13d**). These data reinforce the validity of BF-Sparse-SIM in reporting quantitatively accurate kinetics.*

My comment: *The author calculated the FRC numbers as suggested. However, I didn't see improved image quality with the BF procedure from these values.*

Reply: *Thank you. In comparison to Wavelets-Sparse-SIM, BF-Sparse-SIM does not offer improved resolution since resolution enhancement relies on sparsity, and BF is a technique used for background removal. It's important to note that BF-Sparse-SIM not only improves resolution but also enhances signal integrity. While resolution is critical for*

live cell imaging, maintaining signal fidelity is equally essential.

4. I would also like to note that I disagree with the data availability statement "upon reasonable request". I prefer fully open data, or a statement without this questionable disclaimer of "reasonable".

Reply 8: *Agreed. We have uploaded raw images and reconstructed SR images from **Fig. 1, 2e, f,** and **Supplementary Fig. 3b,c, 6, 9, 10** at <https://doi.org/10.6084/m9.figshare.21792788>. In addition, we have also provided individual example datasets from **Figs. 3, 4,** and **Supplementary Fig. 12** there.*

My comment: Thanks for the open source data.

Reply: *Thank you.*

Finally, I cannot assess the credibility of the novelty claims made on actin dynamics, as I am no expert in actin biophysics.

Reply 9: *As elaborated in **Reply 6**, we have shown three types of actin dynamics with two independent SR methods, confirming their existence in RAW264.7 cells. In addition, we did a literature search in PubMed for actin dynamics in macrophages and other cell types and failed to find any similar actin dynamics reported previously. Therefore, we have revised the statement in the manuscript as "**To our best knowledge,**".*

*We have incorporated parts of the discussion above in the **manuscript (Page 8, Line 23)**.*

My comment: OK.

Reply: *Thank you.*